# Selective semihydrogenation of acetylene in ethylene using defect-rich boron nitride catalyst from flux reconstruction

Tao Wang [1,11], Kevin M. Siniard[2,11], Meijia Li [1], Felipe Polo-Garzon [1], Jue Liu [3] ✉, Zengqing Zhuo [4], Jinghua Guo [4], Alexander S. Ivanov [1,5], Takeshi Kobayashi[6], Kui Tan[7], Stella Amagbor [7], Abdullah Ali Maruf [7], Jeffry Kelber[7], Shize Yang [8], Haohong Song [9], De-en Jiang [9], Gerd Duscher [10], Zhenzhen Yang [1] ✉ & Sheng Dai [1,2] ✉

Efficient removal of trace acetylene from ethylene streams is essential for producing polymer-grade ethylene, yet achieving highly selective semihydrogenation without over-hydrogenation remains a long-standing challenge. A key barrier is the lack of a simple, low-cost catalyst that can activate hydrogen effectively while preventing ethylene from reacting further. Here we show that defect-rich boron nitride, prepared through a straightforward flux reconstruction method, serves as a highly selective and metal-free catalyst for acetylene semihydrogenation. The catalyst contains abundant open boron and nitrogen sites that enable efficient hydrogen activation and rapid release of ethylene, thereby avoiding over-hydrogenation. Experiments combined with isotope labeling and theoretical analysis reveal that these defects lower the energy barrier for hydrogen activation while accelerating product desorption. Our findings demonstrate a scalable strategy for defect engineering in boron nitride and highlight its potential as a robust, sustainable alternative to metal-based catalysts in industrial ethylene purification.

Ethylene is one of the major fundamental building blocks in petrochemical industry to produce plastics, antifreeze solutions, and solvents, in which the purity of the ethylene stream plays critical roles in the following catalytic valorization processes[1–3]. The current ethylene production is primarily through the steam cracking of hydrocarbons, with co-produced acetylene and $H_2$ impurities[4]. Particularly, the presence of acetylene can lead to catalyst poison (e.g., Ziegler-Natta catalysts in polymerization)[5], defect formation in polymer chain, and safety risk in the downstream ethylene conversion[6]. Selective

semihydrogenation of the acetylene-to-ethylene is recognized as an efficient approach to improve the quality of ethylene sources, and the key success factor lies in highly efficient and selective catalytic system design without over-hydrogenation to ethane[7]. State-of-the-art catalysts in selective acetylene hydrogenation mainly involve noble- and transition-metal nanoparticles (NPs), alloys, or single atoms (SAs, e.g., Pd, Au, Cu, and Ni)[8–10], with the active sites being isolated via metal surface modification[11], strong metal-support interaction (SMSI) construction[12], and confinement effect introduction by porosity

[1]Chemical Sciences Division, Oak Ridge National Laboratory, Oak Ridge, TN 37831, USA. [2]Department of Chemistry, Institute for Advanced Materials and Manufacturing, University of Tennessee, Knoxville, Tennessee 37996, USA. [3]Neutron Scattering Division, Oak Ridge National Laboratory, Oak Ridge, TN 37831, USA. [4]Advanced Light Source, Lawrence Berkeley National Laboratory, Berkeley, CA 94720, USA. [5]Department of Nuclear Engineering, University of Tennessee, Knoxville 37996, USA. [6]U.S. DOE Ames National Laboratory, Ames, Iowa 50011, USA. [7]Department of Chemistry, University of North Texas, Denton, Texas 76201, USA. [8]Aberration Corrected Electron Microscopy Core, Yale University, New Haven, CT 06516, USA. [9]Department of Chemical and Biomolecular Engineering, Vanderbilt University, Nashville, TN 37235, USA. [10]Department of Materials Science and Engineering, University of Tennessee, Knoxville, TN 37916, USA. [11]These authors contributed equally: Tao Wang, Kevin M. Siniard. ✉e-mail: liuj1@ornl.gov; yangz3@ornl.gov; dais@ornl.gov

control to ensure high selectivity towards ethylene production[3,13,14]. Despite significant progress, the development of metal-free catalysts remains highly attractive due to the limited natural abundance of platinum group metals (PGMs) and the persistent challenges of sintering and coking in metal-catalyzed hydrogenation processes. The study on $CeO_2$ or $In_2O_3$ catalysts demonstrated the possibility to adopt metal oxide in acetylene semihydrogenation, but inferior ethylene selectivity (< 81%), oligomer by-product formation, and deactivation induced by the redox property of Ce sites limited the practical application[15,16]. The development of metal-free catalytic materials for highly selective acetylene-to-ethylene conversion is appealing, as it addresses the problem of over-hydrogenation and mitigates performance degradation caused by metal sintering—especially in the removal of trace acetylene impurities from ethylene streams. The critical point lies in creating abundant active sites to activate/dissociate $H_2$, facilitate C (from acetylene)-H bond formation, and accelerate ethylene releasing.

Two-dimensional (2D) materials, including transition metal dichalcogenides[17], MXenes[18], hexagonal boron nitride (h-BN)[19,20], and black phosphorus[21,22], have gained ever increasing attention in electronics, energy storage, and catalysis fields benefiting from the development of controllable synthesis and structure engineering technologies[23,24]. Among these, the layered h-BN scaffolds composed of alternative covalent B-N bonds are appealing in terms of high thermal and chemical stability, which has demonstrated the capability to promote catalytic procedures at temperatures up to 800 °C and under harsh chemical environment such as highly corrosive and oxidizing/reducing conditions[19,20,25,26]. Controlling the nanostructure of h-BN offers a venue for advanced electronics and catalysis applications[27-35]. For example, in perfect $sp^2$-hybridized B−N hexagonal units, the binding energies of $H_2$ molecules were calculated to be −0.085 eV and −0.1 eV at the B and N sites, respectively[36]. Theoretical simulations indicate that BN scaffolds with vacancies deviating from the ideal $B_3N_3$ structure are can activate and dissociate $H_2$ molecules through the formation of N−H and B−H bonds[37-40], with the adsorption energy of hydrogen atoms as low as −9.40 eV, depending on the vacancy size and atomic configuration (Supplementary Table 1).

In addition, the physical, chemical, and electronic properties of h-BN can be micro-engineered through tuning crystallinity, exfoliation, defect creation, dopants addition, and heteroatom incorporation[41]. Notably, the high structural tunability of h-BN enables the controlled formation of unsaturated boron and nitrogen sites in close proximity, which can act as strong acid−base centers to activate small gas molecules via frustrated Lewis pair (FLP)-like behavior (Tables S2). Similar mechanisms have been demonstrated in previous studies involving h-BN and B- or N-doped carbon materials[39,42-44]. In addition to FLP-catalyzed hydrogenation, Nash et al. proposed that hydrogenation on defect-rich BN more closely follows the Horiuti−Polanyi mechanism[45] This work pioneered the heterogeneous metal-free hydrogenation over defect-laden h-BN. Furthermore, the activation of small gas molecules via coexisting acid and base sites has also been reported in FLP catalysts based on heteroatom-doped carbon scaffolds[46,47], $CeO_2$[48], and InOH[49], as well as in semi-heterogeneous[50,51] and homogeneous systems[52,53]. In addition, the electronic property of the h-BN scaffolds could be harnessed to tune the adsorption strength of the acetylene substrate and ethylene product, which is critical to achieve highly selective semihydrogenation[3]. Therefore, h-BN catalysts hold great promise for achieving selective acetylene semihydrogenation, with the key challenge being the development of efficient synthesis methods to produce high-quality materials enriched with accessible Lewis acid and base sites.

In this study, the highly selective semihydrogenation of acetylene-to-ethylene and low concentration acetylene impurity removal from the ethylene stream was achieved using defect-rich h-BN catalyst from flux reconstruction. Molten $NaNH_2$ served as a flux medium, nitrogen source, and catalyst to transform amorphous BN (AMBN) architectures into crystalline h-BN nanocrystals (BN-700) through thermal treatment, during which abundant open B and N sites were simultaneously generated as carbon and oxygen were removed (Fig. 1a). The BN structure evolution and chemical bonding variation of the scaffolds upon flux reconstruction was monitored by microscopy, X-ray pair distribution function (PDF), and solid-state $^{11}B$ and $^{15}N$ nuclear magnetic resonance (NMR) combing isotope labelling techniques. The bonding type of B and N within the vacancy sites and the stacking

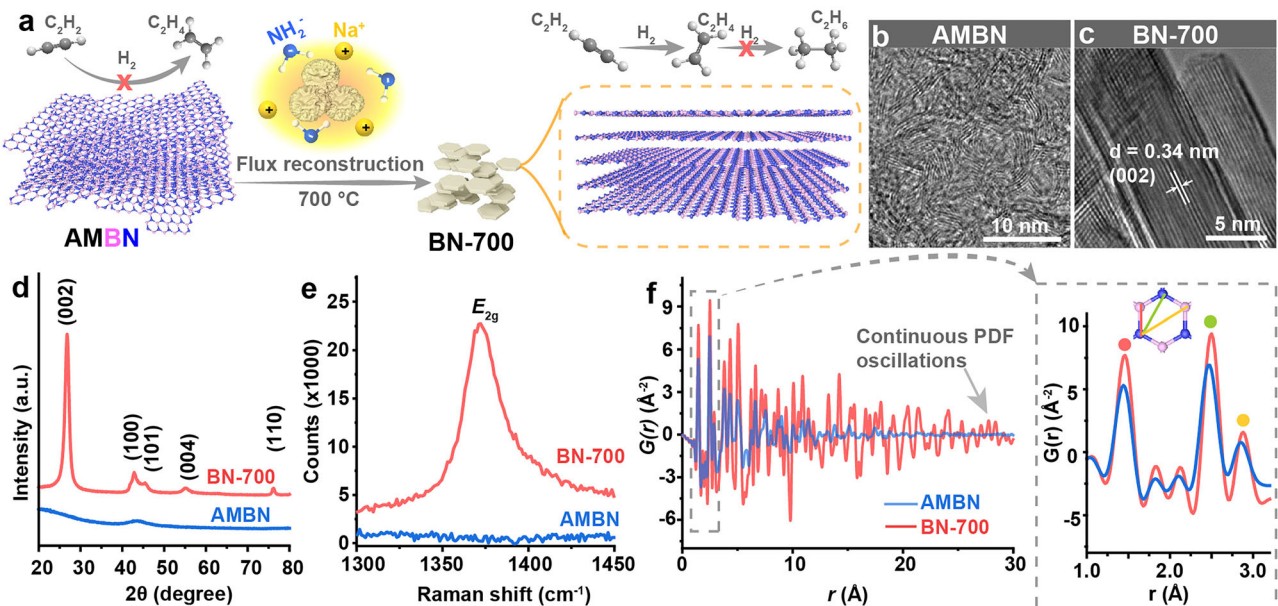

**Fig. 1 | Structure changes before and after flux reconstruction. a** Illustration of the flux reconstruction process from AMBN to BN-700 (Hydrogen, carbon, boron, and nitrogen atoms are represented by white, black, pink, and blue spheres, respectively); **b** TEM image of AMBN; **c** TEM image of BN-700; **d** XRD patterns of AMBN and BN-700; **e** Raman spectra of AMBN and BN-700; **f** Measured atomic pair distribution functions, G(r), of AMBN and BN-700 and a magnified region of G(r) covering short bond distances in the $B_3N_3$ ring. Source data are provided as a Source Data file.

mode in BN-700 was illustrated via near-edge B-*K* and N-*K* X-ray absorption spectroscopy (XAS), together with the neutron diffraction and pair distribution function (PDF) using $^{11}$B-enriched BN-700. The high quality and defect-abundant nature of the BN-700 catalyst rendered it with the capability in $H_2$ activation and dissociation in both liquid (through $H_2/D_2$ exchanging experiment via NMR) and gaseous phase (through $H_2/D_2$ infrared spectroscopy and outlet gas monitoring). The BN-700 catalyst exhibited promising performance in semi-hydrogenation of acetylene, affording >98% selectivity to ethylene even under ethylene-rich gas stream. Fundamental insights from combined isotope effect study, neutron PDF analysis, and theoretical simulation revealed that the C (from acetylene)-H bond formation and B-H bond cleavage as the rate-determined step, and the critical role of defects in h-BN to facilitate ethylene releasing from the catalyst surface.

## Results

### Flux reconstruction and characterization of BN catalyst

Traditional h-BN synthesis methods are not suitable for large-scale fabrication and dedicated structure engineering, as harsh conditions are required, such as high nitrogen pressure (5–15 GPa) using a laser in a diamond anvil cell and high temperatures (from 1100 to 2700 °C) or chemical vapor deposition[54]. Pyrolysis treatment of organic raw materials containing B and N (e.g., urea, borax, boric acid, and melamine) could afford h-BN with large scale but relatively low crystallinity or amorphous nature (denoted as AMBN)[55–57]. AMBN is not catalytically active in hydrogenation reactions due to the quenching effect caused by undesirable carbon and oxygen impurities from the precursors[58–61]. Crystallization and purification of AMBN in the presence of metal catalysts (e.g., Mg) was demonstrated as a promising methodology to fabricate h-BN with improved crystallinity upon thermal treatment at 900 °C[60,62]. However, like most metal-catalyzed graphitization processes[62], besides the requirement of excess metal catalysts, the encapsulated metal moieties were difficult to completely remove[63,64]. Besides metal-involved systems, molten salts have been utilized as a highly charged flux media to synthesize high-temperature phases under relatively low temperature[65,66]. Considering the low melting point (210 °C), strong reducibility, and capability to participate in B-N bond formation, the use of $NaNH_2$ holds great potential for achieving the transformation and purification of AMBN to h-BN nanocrystals at low temperatures without inducing other impurity elements[39,41,67].

To test the proposed synthesis approach, the AMBN precursor was synthesized by the pyrolysis of a boric acid and urea mixture[56]. Then the reconstruction was performed by treating the AMBN and $NaNH_2$ mixture at 700 °C, and the product was collected after washing and drying procedure (denoted as BN-700, Fig. 1a). The morphology and microstructure of AMBN and BN-700 were characterized by scanning electron microscopy (SEM) and transmission electron microscope (TEM). The SEM and low magnification TEM images (Supplementary Fig. 1) of AMBN revealed some 2D features including a micro-sponge like morphology and randomly curled thin sheets. AMBN was proved to be amorphous as few lattices can be found in the high-resolution TEM (HRTEM, Fig. 1b) image and selected area electron diffraction (SAED) pattern (Supplementary Fig. 1c). After flux reconstruction, BN-700 exhibited stacked nanoparticle morphology in SEM image (Supplementary Fig. 1e) and clear lattice patterns corresponding to (002) planes of h-BN in HRTEM image (Fig. 1c). The surface area of BN-700 was calculated to be 160 $m^2 g^{-1}$ based on the $N_2$ adsorption and desorption isotherms being collected at 77 K, which exhibited a steep hysteresis at higher relative pressure above 0.6, indicating the existence of mesopores from the stacking of h-BN nanocrystals (Supplementary Fig. 2).

The crystalline phases of AMBN and BN-700 were identified by powder X-ray diffraction (PXRD, Fig. 1d). Only a broad peak at 43.5° was found in the XRD pattern of AMBN, validated its amorphous structure. After flux reconstruction in $NaNH_2$, the as-collected BN-700 exhibited a sharp diffraction peak at 26.7° being assigned to the (002) crystal plane (assuming the ABAB… stacking registration of BN layers) of the hexagonal architecture. In addition, diffraction peaks from (100), (101), (004), and (110) crystal planes of h-BN were identified in the PXRD pattern of BN-700, indicating the highly crystalline structure. The average crystallite size (D), microstrain (ε), and dislocation density (δ) were determined from the XRD pattern of BN-700 using the uniform deformation model (UDM)[68], based on the Williamson-Hall (W-H) and Williamson-Smallman (W-S) methods, after correcting for instrumental broadening with a silicon standard. This analysis assumes uniform strain across all crystallographic directions. The W-H plot and correction details are shown in Supplementary Fig. 3. The crystallite size was estimated to be 5.8 nm. The negative strain value (ε = −0.0023) indicates compressive strain, likely due to lattice defects or dislocations, as further supported by the high dislocation density (δ = 2.9 × 10$^{16}$ m$^{-2}$). Raman spectra as a powerful tool to investigate the structure of 2D materials was used to identify the existence of typical h-BN features. In Raman spectra (Fig. 1e), BN-700 exhibited a strong $E_{2g}$ vibration band at about 1370 cm$^{-1}$, corresponding to the stacked hexagonal structure of h-BN[69], which was absent in AMBN. Additionally, to gain atomic scale insights into the local structure and defect formation in the obtained AMBN and BN-700 materials, high-energy X-ray scattering measurements were performed at the Advanced Photon Source (APS) synchrotron facility utilizing a pair distribution function (PDF) approach[70]. Unlike the common XRD technique, total X-ray scattering data contain both Bragg scattering and diffuse scattering in the corresponding structure functions (Supplementary Fig. 4), which were Fourier transformed into the real-space PDFs, $G(r)$, showing all atomic pair distances in the studied systems. The $G(r)$ patterns in Fig. 1f indicate that both AMBN and BN-700 exhibit similar local structural correlations, resembling that of the planar graphitic arrangement of atoms[71,72]. In particular, the first three most intense peaks at ~1.46, 2.48, and 2.88 Å signify the in-plane bond distances in the BN six-membered rings. The peak intensities of BN-700 are noticeably higher than that of AMBN, suggesting very limited coherence of BN domains. In addition, integrating the first peak in the PDFs with appropriate X-ray weighting factors leads to the nearest-neighbor coordination numbers of 2.4 for BN-700. Therefore, despite the high crystallinity, BN-700 still preserves the appearance of local unsaturated B- and N-sites in the scaffold, since the determined average first coordination number of 2.4 deviates from an ideal value of 3 in the perfect periodic h-BN structure. The maximum distance at which peaks are observable in PDF gives insight to the size of coherent domain. As shown in Fig. 1f, AMBN represents an amorphous material with the PDF signal damping already at relatively short distances ( ~ 12 Å), while BN-700 shows continuous PDF oscillations at much longer distances (beyond 25 Å), indicating the presence of larger crystalline domains with the size of 5 nm in its structure.

Along with the crystallization, the chemical structure and bonding modes variation during the flux reconstruction was studied using solid-state nuclear magnetic resonance (SS-NMR) analysis. The magic angle spinning (MAS) $^{11}$B SS-NMR spectrum of AMBN displayed the presence of $BO_3$ (−0.7 ppm) and $BN_3$ (10.1 and 21.8 ppm) units within the skeleton (Supplementary Fig. 5). Comparatively, BN-700 being obtained after the flux reconstruction exhibited improved purity, with characteristic peaks assigned to the $BN_3$ units well-maintained while the absence of O-containing moieties (Supplementary Fig. 5). The B-bonding variation was further scrutinized by the $^{11}$B triple-quantum MAS experiments. The AMBN precursor displayed the presence of tetracoordinated $BN_xO_{4-x}$ (x = 0-4) species, together with the regular tricoordinate $BN_3$ bonds (Fig. 2a and Supplementary Fig. 6). In contrast, the content of the tricoordinate $BN_3$ species was significantly increased while the O-involved impurities were almost invisible in BN-700 (Fig. 2b). Detecting the transformation of N-bonds by NMR

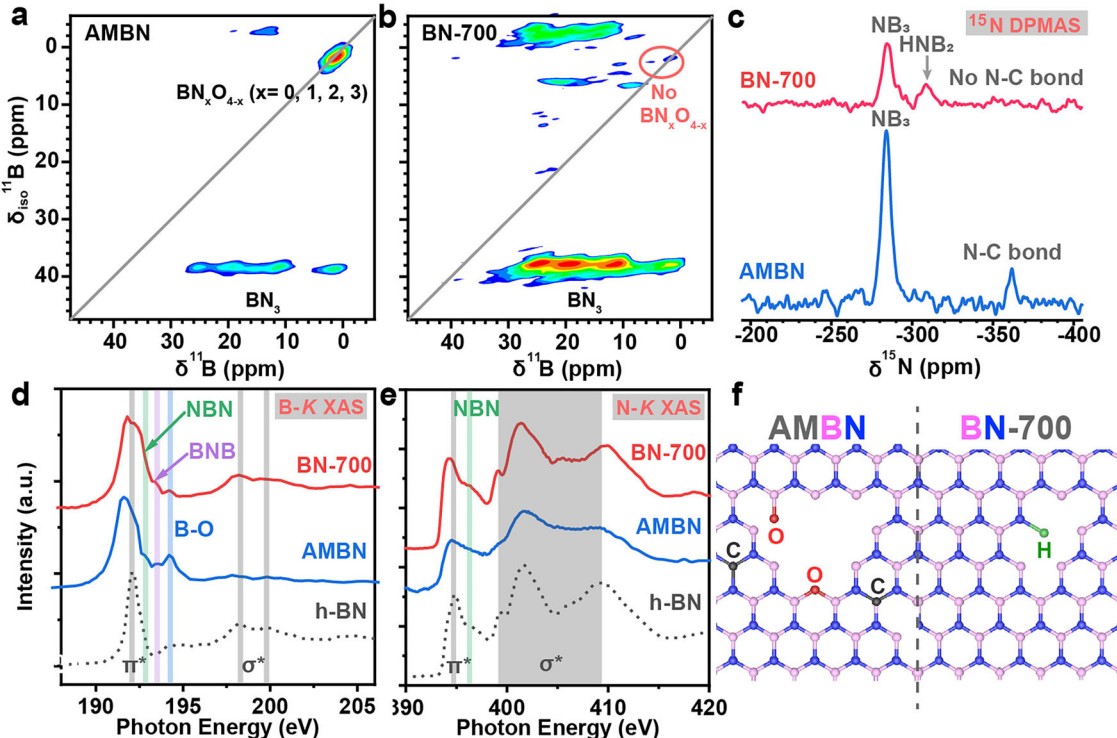

**Fig. 2 | Bonding changes before and after flux reconstruction. a, b** $^{11}$B triple-quantum MAS NMR spectra of AMBN and BN-700. Signals above the diagonal line are spinning sidebands, noise, or artifacts. **c** $^{15}$N DP SS NMR spectra of $^{15}$N-labeled AMBN and the as-afforded BN-700 via flux reconstruction. Only $^{15}$N signals in BN-700 were shown up. **d, e** B-$K$ and N-$K$ XAS spectra of the AMBN and BN-700 samples, with spectra of standard sample h-BN shown as dotted line in the bottom. **f** Illustration of bonding structures of AMBN and BN-700. Hydrogen, oxygen, carbon, boron, and nitrogen atoms are represented by green, red, black, pink, and blue spheres, respectively. Source data are provided as a Source Data file.

technique was more challenging considering the low natural abundance of $^{15}$N ($\sim$0.4 %). Therefore, a $^{15}$N-enriched AMBN sample was prepared with $^{15}$N-labeled urea as the starting precursor. The flux reconstruction was then performed using regular NaNH$_2$. The SS $^{15}$N NMR analysis will provide information on the N-bonds only related to the $^{15}$N from the AMBN precursor. As shown in Fig. 2c, in the SS $^{15}$N NMR spectrum of AMBN, besides the peak of $^{15}$NB$_3$ structure, the residual carbon impurities led to the formation of $^{15}$N-C bonds. Comparatively, after NaNH$_2$ treatment, the $^{15}$NB$_3$ units were maintained in the as-afforded BN-700 product, while the $^{15}$N-C signal disappeared, which in turn, led to the formation of $^{15}$N-H bond in BN-700. Thermogravimetric-mass spectrometry (TG-MS) tests were conducted to analyze the gas products released during flux reconstruction. As shown in Supplementary Fig. 7, the weight loss of the AMBN/NaNH$_2$ mixture around 200 °C is primarily due to the release of NH$_3$, H$_2$, and H$_2$O. As the temperature increases to 700 °C, the MS signals of CO$_2$ become more pronounced, indicating its release at higher temperatures. Thus, the main gaseous byproducts during flux reconstruction are NH$_3$, H$_2$O, H$_2$, and CO$_2$. The O and C impurities in AMBN are converted into H$_2$O and CO$_2$.

The creation of open B and N sites and the existence of vacancy in BN-700 was characterized by soft x-ray spectroscopy. B-$K$ and N-$K$ X-ray absorption spectroscopy (XAS) of AMBN and BN-700 were displayed in Figs. 2d, e. Generally, the dominating main peak feature at 192 eV stems from B 1 s to $\pi^*$ transition, followed by a broader $\sigma^*$ region at 6-8 eV higher energy, which displayed in both AMBN and the BN-700 samples (Fig. 2d). Additionally, a strong B-O feature at 194.2 eV was observed in AMBN, which was largely diminished in BN-700, indicating cleavage of most B-O bond during the flux reconstruction procedure. The surface oxygen content decreased from 14.9 at.% in AMBN to 3.6 at.% in BN-700, as determined by X-ray photoelectron spectroscopy (XPS, Supplementary Fig. 8). Prominently, satellite features

located at 0.7 and 1.4 eV higher than the main peak in BN-700 was observed (Fig. 2d), which was attributed to the N-B-N and B-N-B bonding features respectively with open B sites derived from the cleavage of B-O bonds in the AMBN precursor. The N-$K$ XAS spectra of AMBN and BN-700 exhibited typical h-BN $\pi^*$ features located at 401.7 eV and a broad $\sigma^*$ region (Fig. 2e), which was consistent with the B-$K$ XAS spectra. A shoulder peak at 396.2 eV was shown up in BN-700 illustrating the presence of N-B-N defects. Notably, the features in N-$K$ XAS spectra of BN-700 were sharper than the AMBN precursor, indicating the improved crystallinity. High-resolution site-specific high-angle annular dark-field (HAADF-STEM) images together with electron energy loss spectroscopy (EELS) mapping were captured from various microscopic regions of BN-700, which revealed the existence of abundant B- and N-defects confirmed by the non-uniform distribution of B and N elements at sub-angstrom spatial level (Supplementary Fig. 9).

### Determine defect structure in BN-700 via neutron PDF

Neutron diffraction and PDF data were collected to characterize the defect structure and stacking sequence of BN layers in BN-700, which possesses higher sensitivity to light elements such as B and N. Since natural B is a strong neutron absorber, a pure $^{11}$B-enriched boric acid was deployed as the starting materials to synthesize $^{11}$B-enriched AMBN and crystalline$^{11}$BN-700 to provide clear picture of defect chemistry, short-range B bonding environments and corresponding stacking registration in these Van der Waals materials[73]. First, reduced neutron PDF data of AM$^{11}$BN and $^{11}$BN-700 were collected and compared (Fig. 3a). The negative peak around 1 Å in the AM$^{11}$BN sample is from surface terminated $^{11}$B-H or N-H bonds, which were absent in $^{11}$BN-700 after flux reconstruction[74]. The shoulder peak (adjacent to the B-N bond) around 1.5 Å is likely associated with the partial formation of the $SP^3$ hybridized B bonds, which may be caused by the in-plane N defects

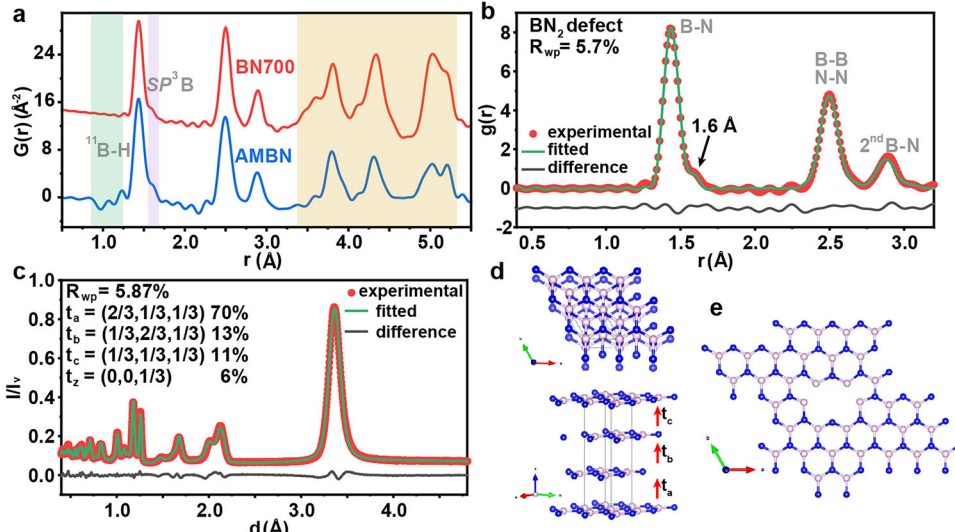

**Fig. 3 | Defects and stacking structures of AM$^{11}$BN and $^{11}$BN-700. a** The neutron PDF data of AM$^{11}$BN and $^{11}$BN-700. **b** The PDF pattern of BN-700 with modeling-derived curves for BN model with BN$_2$ defect structure (off-plane relaxation was allowed). **c** The fitting results of the stacking model composed of stacking probability of 70% $t_a$, 13% $t_b$, 11% $t_c$ and 6% $t_z$. A total of 200 layers with 100 sequences (20000 layers in total) was used during the simulation. Inversion center for Aa... stacking was created by inducing two different BN layers. **d** Illustration of stacking mode. **e** Illustration of BN2 defect structure. Boron and nitrogen atoms are represented by pink and blue spheres, respectively. Source data are provided as a Source Data file.

or surface termination defects. The peak intensity of this shoulder peak in $^{11}$BN-700 decreased compared with the AM$^{11}$BN precursor, suggesting the partial conversion of the $SP^3$ hybridized B to the planar $SP^2$ hybridized B[75]. Noticeable changes were also observed at larger atomic pair distances, e.g., around 3.5 Å and 5 Å, which were likely associated with the evolution of stacking sequences of BN layers along c-axis direction as will be discussed in the latter section. To figure out the defect chemistry associated with the shoulder peak around 1.5 Å and the variations in the longer distance region (3-5 Å), neutron PDF fitting was performed using BN models with different vacancy structures. The previous work revealed that in a h-BN sheet model with a fixed 6×6×1 supercell, six vacancy models can be created with reasonable formation energy and stability, by removing different combination of B and N atoms from the perfect hexagonal unit cell (Supplementary Fig. 10)[39]. The neutron PDF fitting was then conducted based on these established models. First, compared with the perfect h-BN unit cell, it was demonstrated that the first three nearest atomic pairs were from in-plane atomic pair correlations, i.e., the B-N bonds, the nearest B-B and N-N distances, and the second nearest B-N pairs (Supplementary Fig. 11), but leading to a high refinement residual value ($R_{wp} = 9.1\%$) and relatively poor fit. For the h-BN models with different vacancy types and sizes, the fitting results revealed that h-BN with BN$_2$ vacancy provides best fit and lowest refinement residual values, with the trend of BN$_2$ ($R_{wp} = 9.08\%$) <BN$_3$ ($R_{wp} = 10.23\%$) <BN ($R_{wp} = 10.44\%$) <B$_3$N$_3$ ($R_{wp} = 11.55\%$) <B$_7$N$_6$ ($R_{wp} = 13.56\%$) <B$_7$N$_{12}$ ($R_{wp} = 13.73\%$) (Supplementary Fig. 12-17). Further improvement on the fitting results using h-BN model with BN$_2$ defects was attained by allowing off-plane defect relaxation, generating a defected h-BN model with the best fit (Fig. 3b, e). It should be noted that h-BN catalysts may contain a variety of defect types. However, neutron PDF analysis cannot resolve or quantify multiple coexisting defects due to its ensemble-averaged nature and the small free energy differences between various defect configurations.

While neutron PDF provides useful insights regarding the plausible defect chemistries within the BN honeycomb layer. The stacking mode of these BN honeycomb layers in BN-700 was further investigated using combined neutron and X-ray Bragg diffraction data. The most broadly cited stacking model is the bi-layer A-a stacking model (B and N facing each other on the adjacent layers) with the space group of

$P6_3$/mmc (Supplementary Fig. 18). However, for the BN-700 herein, when structure refinement was performed using the AA (B facing B, and N facing N in the successive layers) or Aa (B-N alternative stacking in the successive layers) stacking of B-N layers, the fits were very poor, with a very large refinement residual $R_{wp}$ (>20%, Supplementary Fig. 19). In addition, the fit was even worse when using the AB stacking of B-N layers (similar to that of the graphite, Supplementary Fig. 20). This suggests that the widely reported bi-layer stacking model may not represent the true structure of the currently synthesized BN-700. We then carried out the structure refinements with the ABC tri-layer stacking of B-N layers (Supplementary Fig. 21). There are three plausible translation vectors for the ABC type stacking, among which the $t_at_at_a$ and $t_bt_bt_b$ type stacking lead to the same structure while the $t_ct_ct_c$ type stacking results in a different stacked structure. We found that the pure $t_ct_ct_c$ type stacking resulted in poor quality fit (Supplementary Fig. 21), while much better fit can be achieved with the $t_at_at_a$ type tri-layer stacking (Supplementary Fig. 22). However, there was still noticeable discrepancies in the fit, very likely due to the presence of stacking faults. Diverse hybrid stacking models were tested and good quality of fit can be achieved by constructing numerical stacking faults model. We found that very good quality of fit can be achieved with a mixture stacking probability of 70% $t_a$, 13% $t_b$, 11% $t_c$ and 6% $t_z$. It is worth noting that at this stage no exhausting search was performed to identify the best combination of $t_b$, $t_c$ and $t_z$ stacking in the structure (Fig. 3c, d). Notably, the PXRD pattern of BN-700 can also be well fitted using the stacking faults model derived from the fit of the corresponding neutron diffraction data, with low $R_{wp}$ value (3.85%, Supplementary Fig. 23).

## H$_2$ activation and dissociation capability of BN-700
The facile flux reconstruction approach, high crystallinity, and rich defects within the scaffolds made the as-afforded BN-700 materials promising catalyst in hydrogenation reactions[39,60]. Previous works demonstrated the critical roles of the open B and N sites to activate hydrogen molecules via the FLP manner and adsorb the substrates[39,42–45]. Existence and property of the active sites was evaluated via the temperature-programmed desorption (TPD) technique with CO$_2$ and NH$_3$ as the probe molecule respectively. The CO$_2$-TPD of BN-700 being obtained via NaNH$_2$-assisted flux treatment displayed

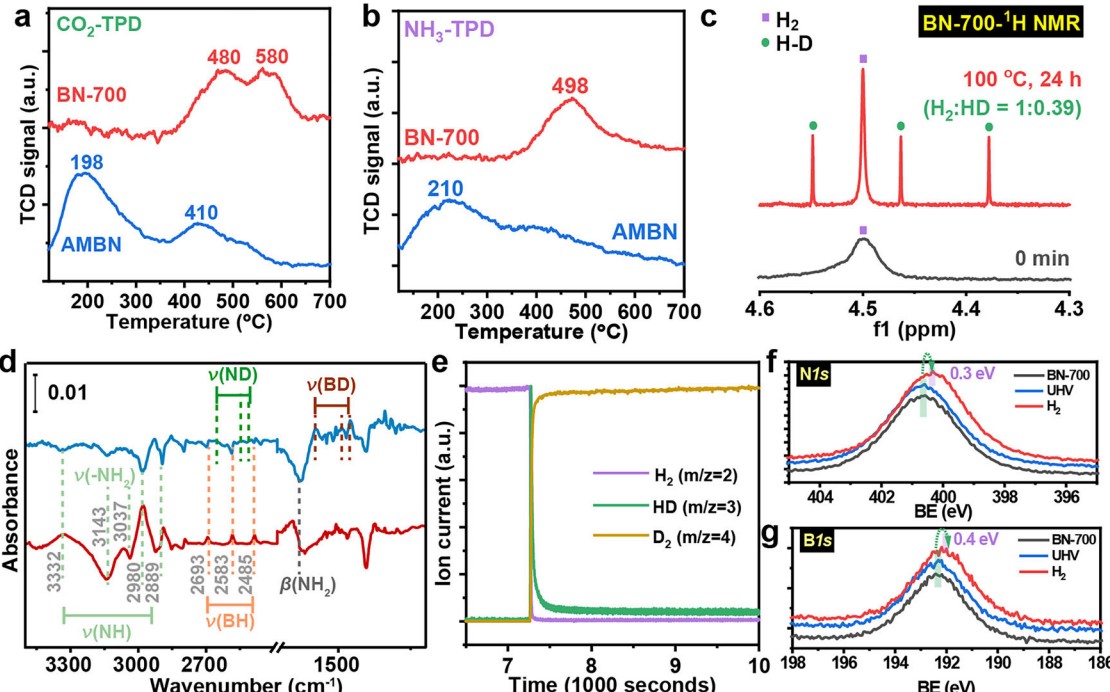

**Fig. 4 | TPD and H₂ dissociation characterizations. a** $CO_2$-TPD **b** $NH_3$-TPD results of AMBN and BN-700; **c** The liquid phase $H_2$-$D_2$ exchange experiments over BN-700 detected by ¹H NMR in $d_{12}$-cyclohexane at 100 °C. **d** The $H_2/D_2$ DRIFTS experiments over BN-700 catalysts being performed at 250 °C. **e** The mass fragments of the outlet gas flow upon switching the feeding gas from $H_2$ to $D_2$ over BN-700 catalyst at 250 °C. **f, g** In situ XPS spectra (N *1s* and B *1s*, respectively) of BN-700 as synthesized, under UHV condition at 250 °C and under $H_2$ atmosphere at 250 °C. BE stands for binding energy. Source data are provided as a Source Data file.

two strong desorption peaks located at 480 and 580 °C, corresponding to the existence of functionalities with moderate and strong basicity, respectively (Fig. 4a)[76,77]. Comparatively, the $CO_2$-TPD of AMBN mainly exhibited the desorption peak of weakly chemisorbed $CO_2$ at 198 °C (e.g., OH species on the surface)[78,79] and a minor one at 410 °C[77,80]. Notably, in $NH_3$-TPD profiles, the peaks with desorption temperature lower than 300 °C are generally ascribed to $NH_3$ molecules being sorbed by Brønsted acid sites. Lewis acid sites-bonded $NH_3$ molecules require higher temperatures (> 300 °C) to be released[42,81]. Accordingly, a prominent desorption peak at 498 °C was shown up over the $NH_3$-TPD profile of BN-700, which was ascribed to unsaturated boron sites ($BN_2$ species), indicating the existence of strong Lewis acid sites (Fig. 4b)[42]. While only weakly adsorbed $NH_3$ molecules existed in AMBN, which was verified by the desorption peak at much lower temperature region (210°C), demonstrating the lack of strong acidic sites within the skeleton of AMBN. The initial TPD evaluation demonstrated the co-existence of relatively strong Lewis acid and base sites within the scaffolds of BN-700, which can act as the active sites for hydrogen adsorption and dissociation.

The capability of BN-700 towards hydrogen activation/dissociation was further explored in liquid and gaseous phase. First, the liquid phase $H_2$ dissociation behavior of BN-700 was studied by H/D isotope scrambling experiments via liquid proton NMR (¹H NMR, Fig. 4c). The reaction was performed using a mixture of $H_2/D_2$ (ca. 1:1 v/v) at ambient pressure as the feeding source. After bubbling the $H_2/D_2$ mixture through the $d_{12}$-cyclohexane solution dispersed with BN-700, the ¹H NMR spectra of the mixture only displayed the signal for $H_2$ at δ = 4.54 ppm as a single peak. Notably, after thermal treatment of the mixture at 100 °C for 24 h, the formation of HD was observed with the characteristic triplet signal, and the molar ratio of $H_2$:HD was calculated to be 1:0.39, indicating the successful H-H/D-D bond cleavage and H-D bond formation promoted by BN-700. Compared with liquid phase $H_2$ activation, the gas-phase procedure is more challenged in the form of gas-solid interaction. In situ diffuse

reflectance infrared Fourier transform spectroscopy (in situ $H_2/D_2$ DRIFTS) was conducted to probe the surface structure evaluation of BN upon thermal treatment under $H_2$ atmosphere (Fig. 4d). The BN-700 catalyst was pretreated in Ar flow (40 mL min⁻¹) at 400 °C to remove the surface impurities, followed by treating with pure $H_2$ and $D_2$ flow at 250 °C, successively. Taking the spectrum of fresh BN-700 as the background, the IR spectra demonstrated that after being treated with $H_2$ at 250 °C for 30 minutes, the formation of N-H (3686 cm⁻¹) and B-H (1004 cm⁻¹) bonds was observed on BN-700[39,44]. In addition, upon exchanging the gas atmosphere to $D_2$ in the reaction chamber, the B-H and N-H bonds disappeared, along with the maintenance of the C-H bonds. The formation of B-D (2480 cm⁻¹) and N-D bonds (3630 and 3540 cm⁻¹) was confirmed by the characteristic peaks, illustrating the $H_2$ dissociation capability of unsaturated B and N sites in BN-700 via reversible B-H/N-H bond formation[82–84] Furthermore, the outlet gas stream of the DRIFTS cell was analyzed by a mass spectrometer during the change of $H_2$ by $D_2$ flow, which displayed the formation of HD (MS = 3) (Fig. 4e), further demonstrating the hydrogen activation/dissociation capability of BN-700 in the gas phase. In situ XPS analysis was performed to further probe the structure evolution of BN-700 under $H_2$ atmosphere at 250 °C. The sample was stable at this temperature as the B1s and N1s spectra exhibited no shift after ultra-high vacuum (UHV) annealing at 250 °C (Fig. 4f, g). However, after further $H_2$ exposure at 7.3 mbar $H_2$ and 250 °C for 30 min, a slight shift in binding energy by 0.4 and 0.3 eV was observed for both B1s and N1s peak, respectively (Fig. 4f, g), which was consistent with the previously observed phenomena in BN treatment under $H_2$ atmosphere[85]. The higher increase in relative intensity at the B1s core level indicates that B defect sites are more reactive than N sites[86]. Inductively coupled-plasma optical emission spectroscopy (ICP-OES) analysis of the BN-700 catalyst confirmed the absence of detectable Ni, with a detection limit of <0.0009 ppm, indicating that the BN-700 scaffold remained free of Ni contamination under the applied synthesis conditions.

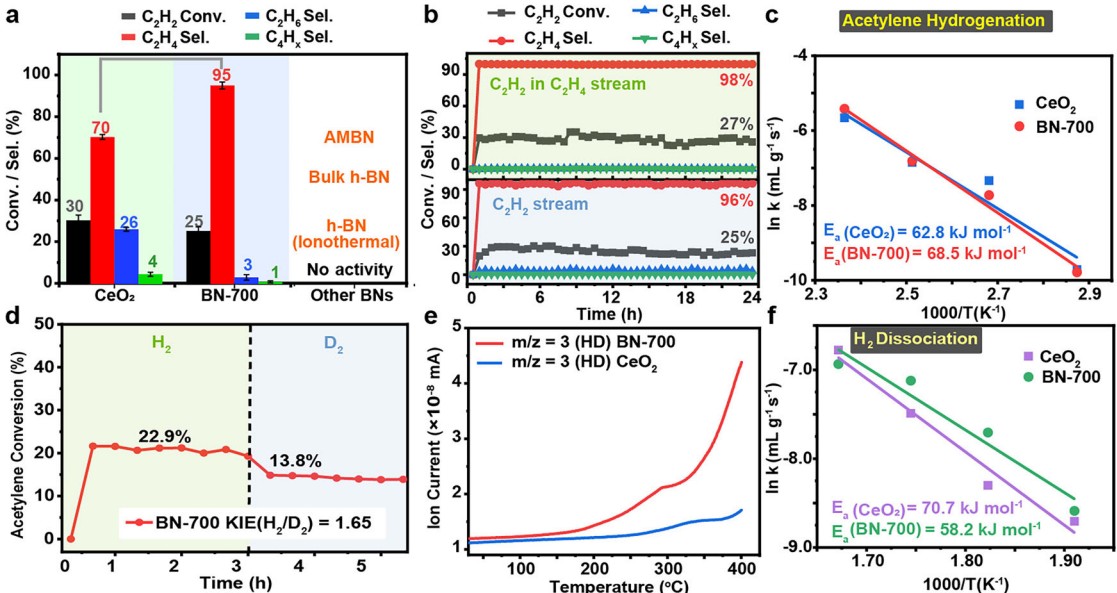

**Fig. 5 | Acetylene semihydrogenation experiments. a** Comparison of the activity of BN-700, CeO$_2$, and other BNs (AMBN, commercial bulk h-BN, and h-BN nanoparticles synthesized through ionothermal method) in acetylene hydrogenation at 150 °C. Error bars stand for standard deviations. **b** Acetylene semihydrogenation (at reaction temperature of 150 °C for 24 h) catalyzed by BN-700 with acetylene in ethylene (up) or only acetylene (bottom) as the feeding gas stream. **c** Arrhenius plots of the acetylene semihydrogenation with the apparent activation energy (E$_a$) for BN-700 and CeO$_2$. **d** KIE measurement of BN-700 in acetylene semihydrogenation in the presence of H$_2$ or D$_2$. **e** HD light-off curves for BN-700 and CeO$_2$. **f** Arrhenius plots of the hydrogen dissociation apparent activation energy (E$_a$) for BN-700 and CeO$_2$. Source data are provided as a Source Data file.

## Acetylene semihydrogenation evaluation and mechanism insights

The unique structural features and hydrogen dissociation activity BN-700 prompted its performance evaluation in acetylene (C$_2$H$_2$) semihydrogenation. The reaction was first performed with a 0.5% C$_2$H$_2$ / 16% H$_2$ / 83.5% Ar mixture over a temperature range of 100 to 150 °C under gas hourly space velocity (GHSV) of 4000 h$^{-1}$ (Supplementary Fig. 24). The results identified BN-700 as an effective metal-free catalyst, with optimal performance observed at 150 °C, achieving a C$_2$H$_2$ conversion of 25% ± 2% and an exceptional C$_2$H$_4$ selectivity of 95% ± 1.6% (Fig. 5a). By-products were minimal, with ethane (C$_2$H$_6$) selectivity below 3% ± 1.3% and oligomerized C$_4$ selectivity below 1% ± 0.5%. The C$_2$H$_4$ selectivity achieved by BN-700 was nearly 1.5 times higher than that of ceria, which has been previously demonstrated as an efficient non-noble metal catalyst for C$_2$H$_2$ semihydrogenation[16,87]. To enhance catalyst performance, variations in catalyst loading and flow rate were systematically investigated to identify optimal conditions for improved activity and ethylene selectivity. Reducing the flow rate to 25 ccm led to an increase in acetylene conversion from 30% to 40%; however, ethylene selectivity dropped from 98% to 81% (Supplementary Fig. 25A). In a separate experiment, increasing the catalyst loading from 75 mg to 300 mg improved conversion from 30% to 46%, but similarly reduced ethylene selectivity to 76% (Supplementary Fig. 25B). For practical acetylene semihydrogenation, the objective is to remove trace acetylene from ethylene streams while preserving high ethylene purity. It is crucial to maintain ethene selectivity above 95%. Therefore, the optimized reaction conditions selected for further studies were: 75 mg catalyst; gas composition: 0.5% C$_2$H$_2$/16% H$_2$/83.5% Ar; flow rate: 50 ccm; GHSV: 4000 h$^{-1}$; and reaction temperature: 150 °C.

Notably, both the AMBN precursor and commercial h-BN, as well as BN-1 (being synthesized from ionothermal procedure)[42], showed no activity under identical conditions, emphasizing distinct catalytic efficiency of BN-700 in selective C$_2$H$_2$ semihydrogenation. Furthermore, BN-700 sustained its conversion and selectivity to C$_2$H$_4$ for 24 h (Fig. 5b, bottom). Notably, the C$_2$H$_4$ selectivity achieved by BN-700 is particularly advantageous in comparison to industrial Pd-alloy catalysts containing precious metals like silver, gold, or copper[88]. In industrial applications, reducing trace acetylene in ethene streams (typically from 1% to below 5 ppm) is essential and highly challenging for maintaining high-purity ethylene in cracking processes[6]. The performance of BN-700 in acetylene removal from ethene-rich stream was tested by deploying a 0.25% C$_2$H/10% C$_2$H$_4$/16% H$_2$/73.5% Ar mixture at 150 °C with a GHSV of 4000 h$^{-1}$. Under these conditions, BN-700 achieved an even higher ethene selectivity of 98% ± 0.5%, with negligible selectivity to ethane and oligomers (<1%) (Fig. 5b, top). Notably, BN-700 maintained its stability in the acetylene/ethylene mixture for 24 h (Fig. 5b, top).

Kinetic and isotopic experiments further underscored the unique catalytic properties of BN-700. As shown in Fig. 5c, BN-700 displayed an apparent activation energy of 68.5 ± 4.6 kJ mol$^{-1}$ for C$_2$H$_2$ hydrogenation, which was comparable to ceria (62.8 ± 3.4 kJ mol$^{-1}$). This led us to further investigate kinetic isotope effects (KIEs). As shown in Fig. 5d, BN-700 exhibited a KIE of 1.65 showing that hydrogen dissociation is an important step in the reaction. Therefore, further H$_2$ dissociation activity was evaluated via isotope (H$_2$ and D$_2$) study and compared with the known ceria catalyst. The HD light-off curves confirmed that BN-700 facilitates more efficient H$_2$ dissociation than ceria under identical conditions (Fig. 5e). Ceria exhibited a higher activation energy (E$_a$ = 70.7 ± 2.5 kJ mol$^{-1}$) for H$_2$ dissociation compared to BN-700 (E$_a$ = 58.2 ± 5.1 kJ mol$^{-1}$) (Fig. 5f), supporting the remarkable H$_2$ activation capabilities of BN-700. This indicated that the defects in BN-700 were highly efficient in terms of H$_2$ dissociation and envisaged us to further explore the energy profile of elementary steps in the reaction procedure.

To better understand why hydrogenation terminates at ethylene formation, we performed in situ DRIFTS analysis during acetylene semi-hydrogenation using a gas mixture of 1% C$_2$H$_2$/Ar and 5% H$_2$/Ar (both at 25 ccm; total flow: 50 ccm). Spectra were collected at 25, 100, and 150 °C using BN-700 as the background reference. At 25 °C, new bands appeared at 3254 and 3318 cm$^{-1}$[87,89–91], attributed to physisorbed

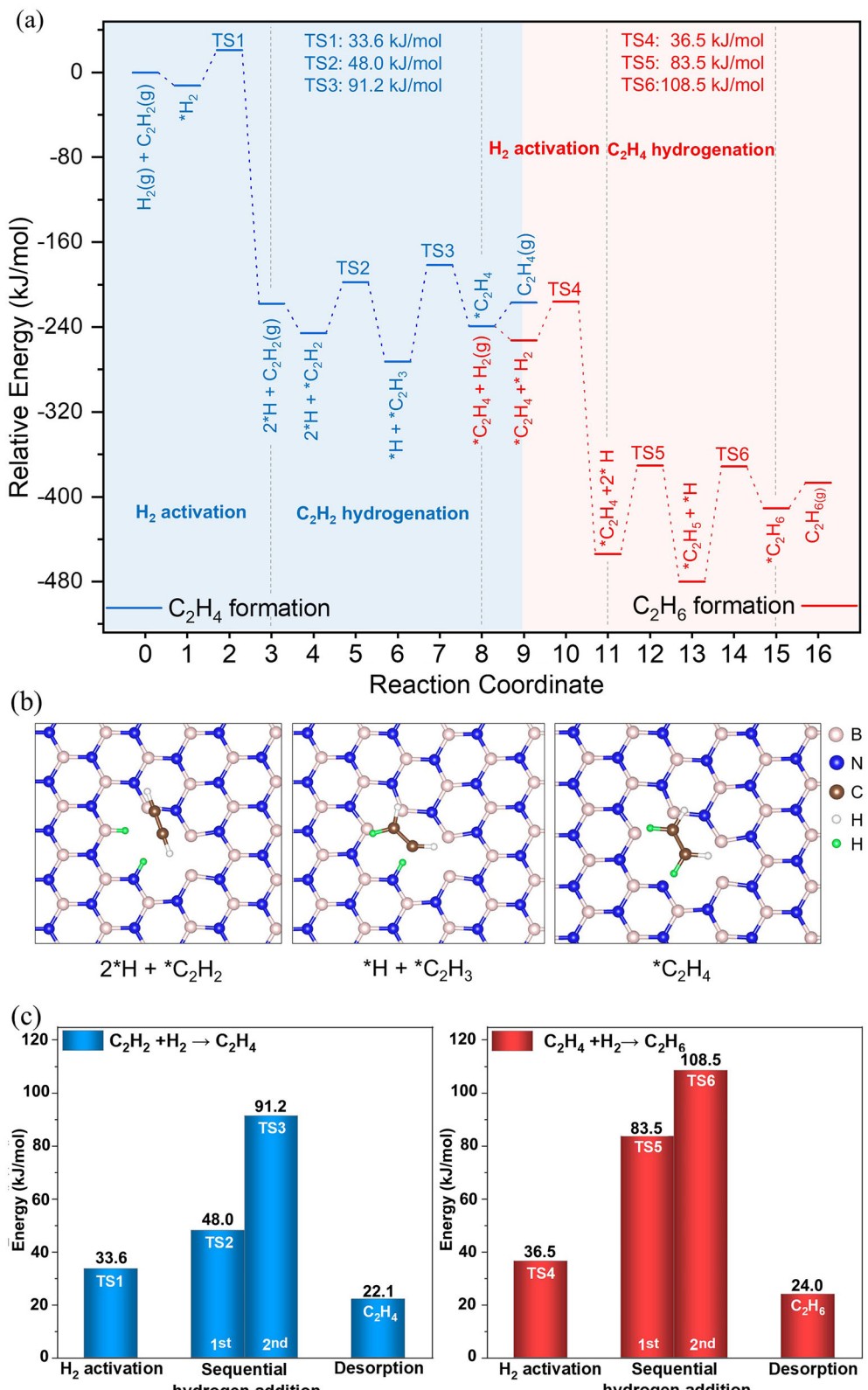

**Fig. 6 | Mechanisms of semi- and full-hydrogenation of acetylene on the BN₂-defected BN catalyst base on DFT modeling. a** DFT-computed energy profiles; **b** structures of activated $H_2$ (green balls) and adsorbed $C_2H_2$ (left), after first hydrogen-atom addition (middle), and after second hydrogen-atom addition (right); Hydrogen, carbon, boron, and nitrogen atoms are represented by white, brown, pink, and blue spheres, respectively. Hydrogen atoms part in the reaction were highlighted as green spheres. **c** Comparison of key energetics between semihydrogenation (left) and full hydrogenation (right) of acetylene. The computational models used for DFT calculation have been supplied as Supplementary Data 1. Source data are provided as a Source Data file.

acetylene molecules, which remained observable upon heating to 100 and 150 °C (Supplementary Fig. 26). Upon increasing the temperature to 100 and 150 °C, additional features emerged. The appearance of N–H stretching bands at 3411 and 3670 cm$^{-1}$[13,87], along with B–H stretching bands at 2498 and 2770 cm$^{-1}$[39,92], confirms that BN-700 effectively activates and dissociates hydrogen (Supplementary Table 3). A notable band at 1202 cm$^{-1}$, assigned to B–N–H vibrations[93], further supports this observation. The persistent signals at 3254 and 3318 cm$^{-1}$ may arise from acetylene adsorption via hydrogen bonding to exposed nitrogen sites[87,89–91]. Importantly, the formation of ethene was detected at elevated temperatures, with its adsorption on the BN surface occurring via weak π-complexation, consistent with a C = C stretching band near 1637, 1756 and 1798 cm$^{-1}$[87,91,94]. Notably, no absorption bands were observed around 2900-3000 cm$^{-1}$[95,96], which would indicate the presence of strongly bound =C–H stretches of ethylene or –CH$_3$ stretches in ethane. This suggests that ethylene, once formed, desorbs readily rather than undergoing further hydrogenation to ethane, thus explaining the selective termination of hydrogenation at the ethylene stage under the tested reaction conditions.

To gain atomistic insights into the high selectivity of the BN-700 catalyst in semihydrogenation of C$_2$H$_2$, we employed the BN$_2$ defect model in Fig. 3e and computed the energy profiles for both the semi- and full-hydrogenation pathways with density functional theory (DFT). The computational models used for DFT calculation have been supplied as Supplementary Data 1. The complete profiles are shown in Fig. 6a. One can see that the H$_2$ activation at the B-N Lewis acid-base pair (Fig. 6b) is both thermodynamically and kinetically favorable, despite that the DFT-computed barrier of 33.6 kJ/mol (TS1 in Fig. 6) underestimates the experimental value of 58.2 kJ/mol (Fig. 5f). First hydrogen atom addition to C$_2$H$_2$ has a slightly higher barrier of 48 kJ/mol (TS2) than the second hydrogen atom addition which is the rate-limiting step with a barrier of 91.2 kJ/mol (TS3). More important, the desorption energy of C$_2$H$_4$ (22.1 kJ/mol) is much lower than the activation energy of C$_2$H$_4$ hydrogenation (83.5 kJ/mol for first H atom addition, TS5; 108.5 kJ/mol for second H atom addition, TS6; Fig. 6c). It should be noted that DFT calculations can yield qualitative insights; however, their accuracy is limited by the approximate description of exchange–correlation interactions and the omission of explicit catalytic environments. DFT computations suggest that the observed high semihydrogenation selectivity of the defected BN catalyst may arise from a combination of factors, such as the potential for B–N pairs to facilitate H$_2$ activation, the presence of moderately high barriers for sequential H-atom additions to C$_2$H$_2$, relatively weak adsorption of C$_2$H$_4$, and comparatively more difficult further H-atom additions to C$_2$H$_4$. Given the challenges in accurately modeling the exact material under catalytic conditions, these results should be regarded as qualitative. On the basis of these tentative insights, together with DRIFTS observations, a schematic catalytic cycle is proposed (Supplementary Fig. 27).

## Discussion

Highly selective acetylene semihydrogenation was achieved using BN as a metal-free catalyst. With pure acetylene or low-concentration acetylene in ethylene stream as the feeding gas, high selectivity towards ethylene production was illustrated surpassing the traditional noble-/transition-metal derived catalysts and metal oxides. The major progress lies in the development of defect-rich BN scaffolds via a flux reconstruction procedure, which exhibited the benefits in creating abundant open B and N sites, affording BN scaffolds with high crystallinity and purity. Isotope-labelling BN scaffolds was deployed to provide insights and structure evolution during the flux reconstruction, combining solid NMR and neutron-based techniques. The existence of abundant Lewis acid and base sites, as revealed by CO$_2$ and NH$_3$ TPD analysis, rendered the BN scaffolds as highly efficient catalyst to activate/dissociate H$_2$ in both liquid and gas phase. The subsequent

semihydrogenation of acetylene in ethylene stream was endowed by the low energy barrier and the acceleration of ethylene releasing from the defect sites.

## Methods

### Chemicals
Urea (99%), $^{15}$N-urea (98 atom % $^{15}$N, 99% CP), boric acid (99.5%), $^{11}$B-boric acid ( ≥ 99 atom % $^{11}$B), CeO$_2$ (98%) and NaNH$_2$ (95%) were purchased from Sigma-Aldrich. Styrene (98%) and Phenylacetylene (98%) were purchased from Alfa Aesar.

### Synthesis of AMBN
The mixture of urea and boric acid with a mass ratio of 5 were physically mixed by hand griding. Then the mixture was placed in an alumina crucible and heated to 900 °C at 10 °C/min under nitrogen gas flow, held at 900 °C for 3 h. A white powder was collected after cooling down to room temperature naturally.

### Synthesis of $^{15}$N-labeled AMBN
The mixture of $^{15}$N-urea and boric acid with a mass ratio of 5 were physically mixed by hand griding. Then the mixture was placed in an alumina crucible and heated to 900 °C at 10 °C/min under nitrogen gas flow, held at 900 °C for 3 h. A white powder was collected after cooling down to room temperature naturally.

### Synthesis of $^{11}$B-labeled AMBN
The mixture of urea and $^{11}$B-boric acid with a mass ratio of 5 were physically mixed by hand griding. Then the mixture was placed in an alumina crucible and heated to 900 °C at 10 °C/min under nitrogen gas flow, held at 900 °C for 3 h. A white powder was collected after cooling down to room temperature naturally.

### Synthesis of BN-700
For the synthesis of BN-700, the mixture of NaNH$_2$ and AMBN with a mass ratio of 2 were physically mixed by hand griding in an Ar-filled glove box. Then the mixture was placed in a nickel crucible and heated to 350 °C at 5 °C/min, held for 1 h, and then to 700 °C at 5 °C/min, held for 2 h, under nitrogen gas flow. A white powder was collected after cooling down to room temperature naturally. Then the powder was washed with water to remove soluble salts and soaked in 3 M hydrochloric acid and heated to 70 °C in an oven for 5 h. Then, the powder was separated from acidic solution and washed with water for three times by centrifugate. Finally, the powder was dried in a vacuum oven at 110 °C.

**Caution.** NaNH$_2$ is highly reactive with water, releasing ammonia gas and heat upon contact. To ensure safe handling, all synthesis procedures involving NaNH$_2$ were conducted under an inert argon atmosphere using a glovebox to avoid moisture exposure. During post-synthesis washing of boron nitride containing trace amounts of NaNH$_2$, it is essential to conduct all steps in a well-functioning fume hood to safely vent toxic ammonia gas released during the reaction. The crucible with sample inside should be dropped into plenty of cold water to minimize the reaction rate and manage heat evolution. Appropriate personal protective equipment (PPE) must be worn, including chemical-resistant gloves (e.g., nitrile), safety goggles or a face shield, a lab coat with long sleeves, and respiratory protection if a fume hood is not available (though this is not recommended).

$^{15}$N-labeled BN-700 was synthesized using NaNH$_2$ and $^{15}$N-labeled AMBN as raw materials and the same synthesis process as BN-700. $^{11}$B-labeled BN-700 was synthesized using NaNH$_2$ and $^{11}$B-labeled AMBN as raw materials and the same synthesis process as BN-700.

### Characterizations
Powder XRD (PXRD) was performed on a PANalytical Empyrean diffractometer, operated at 45 kV and 40 mA (scanning step: 0.02 ° per

step). The diffraction patterns were collected in the range of 20-80°. $\lambda = 0.1540598$ nm. The Williamson-Hall formula ($\beta_{corrected}\cos(\theta) = 4\varepsilon\sin(\theta) + K\lambda/D$) was utilized to determine the average crystallite size (D) and micro-strain ($\varepsilon$) of BN-700 after correcting for instrumental broadening. Instrumental broadening was corrected with a Silicon powder standard where the peak integral breadth ($\beta$ = Peak Area / Peak Intensity) of each peak was utilized for correction according to the following formula: $\beta_{corrected} = \sqrt{(\beta_{BN-700})^2 - (\beta_{Silicon\ Standard})^2}$. The uniform deformation model (UDM) was used to determine the dislocation density ($\delta$) according to the Williamson-Smallman formula ($\delta = 1/D^2$) where it is assumed that that strain is uniformly distributed across all crystallographic directions.

Raman spectra was performed on a Renishaw In-Via Raman spectrum instrument with a 532 nm emission laser.

Magic angle spinning (MAS) [11]B SS-NMR was performed using a solid-state Varian INOVA 400 MHz using densely packed powders.

Thermogravimetric-mass spectrometry (TG-MS) tests were conducted using a Discovery TGA-MS under a nitrogen ($N_2$) atmosphere. The heating protocol involved heating the AMBN/NaNH$_2$ mixture from room temperature to 700 °C at a rate of 5 °C/min, followed by a 30-minute hold at 700 °C.

Elemental composition was determined using an Agilent 5110 inductively coupled plasma–optical emission spectrometer (ICP-OES).

The nitrogen adsorption and desorption isotherms were collected at 77 K on a 3-Flex Micromeritics surface area analyzer. The samples were degassed at 150 °C for 24 h before the measurements.

## Temperature-programmed desorption (TPD)

TPD of $NH_3$ and $CO_2$ was carried out on a Micromeritics AutoChem II 2920 system. About 100 mg of catalyst was placed in a U-shaped quartz reactor, pretreated in He at 350 °C for 1 h, and then cooled to 120 °C under the same atmosphere. Adsorption was performed at 120 °C by passing either $NH_3$ or $CO_2$ (30 mL min$^{-1}$) for 30 min, with He (30 mL min$^{-1}$) as the carrier gas. Physically adsorbed species were removed by purging with He at 120 °C for an additional 30 min. TPD profiles were then recorded from 120 to 700 °C at a ramping rate of 10 °C min$^{-1}$ in He, and desorption was detected using a TCD.

*Kinetic H/D exchange measurement by liquid NMR* was performed by adding 0.1 g BN-700 and 0.5 mL d$^{12}$- cyclohexane in the Young-type NMR tube, which was bubbling with $H_2/D_2$ mixture (1:1, v/v) for 15 min, and sealed. The [1]H NMR spectrum for 0 min was collected without other treatments. The sealed tube was put into a pre-heated oil bath (100 °C) for a certain time (12 h, 24 h, or 48 h), and cooled down to room temperature before collecting the [1]H NMR spectra.

## In situ infrared spectroscopy (IR)

DRIFTS measurements of $H_2$ chemisorption were performed on a Thermo Nicolet Nexus 670 spectrometer using a Pike Technologies HC-900 cell (6 cm$^3$). The outlet stream was monitored by a quadrupole mass spectrometer (OmniStar GSD-301 O2, Pfeiffer Vacuum). ~30 mg of sample was pretreated in Ar (30 mL/min) at 400 °C for 0.5 h. Background spectra were collected under Ar at 250 °C, then spectra were recorded after treatment with $H_2/Ar$ (10/20 mL/min) at the same temperatures for 0.5 h. Finally, $D_2/Ar$ (10/20 mL/min) was introduced at 250 °C, and spectra were collected after 0.5 and 1 h.

## Microscopy, SEM, TEM, scattering

TEM images were collected on a Zeiss Libra 200 MC. SEM images were collected on a Zeiss Auriga SEM. Transmission electron microscopy (TEM) was conducted on an aberration-corrected FEI Titan S 80-300.

## XPS experiment

BN-700 powdered samples were made into slurry form by mixing with a small amount of methanol, mounted on a tungsten substrate and dried in air. The XPS spectra were acquired with the analyzer in constant pass energy mode, with a pass energy of 50 eV, using unmonochromatized Al K$\alpha$ radiation. The Measurements were carried out with the sample aligned normal to the analyzer axis (normal emission). All XPS data were calibrated to C1s binding energy of 284.8 eV.

## Soft X-ray absorption spectroscopy (XAS)

B-*K* XAS measurement was performed at Beamline 8.0.1 of the Advanced Light Source (ALS) at Lawrence Berkeley National Laboratory (LBNL). N-*K* XAS were collected at Beamline 7.3.1 of the ALS. Both B-*K* and N-*K* XAS measurements were collected at room temperature. All the spectra were normalized to the beam flux measured by the upstream gold mesh.

## High-energy x-ray total scattering measurements

Total scattering measurements were carried out at the 11-ID-B beamline of the Advanced Photon Source (APS). A X-ray beam with a photon energy of 58.6 keV ($\lambda = 0.2116$ Å) and dimensions of $0.5 \times 0.5$ mm$^2$ was used, with each sample measured for 10 minutes. The boron nitride powders were sealed in polyimide (Kapton) capillaries and examined in transmission geometry at ambient temperature using a Perkin Elmer XRD1621 amorphous silicon detector[97–99]. To enable background subtraction, scattering from an empty polyimide tube was also recorded under identical conditions. Detector calibration and sample-to-detector distance alignment were performed with $CeO_2$ as a standard, processed in GSAS-II[98]. Radial integration of the raw patterns yielded Q-space data, where masking was applied to exclude artifacts. Structure functions, S(Q), shown in Supplementary Fig. 4, were generated with *PDFgetX2* by subtracting container scattering, incorporating sample composition, and applying conventional area-detector corrections[97,99]. Real-space pair distribution functions, G(r), were then obtained from Fourier sine transforms of S(Q) using a $Q_{max}$ cutoff of 21.0 Å$^{-1}$:

$$G(r) = \frac{2}{\pi} \int_{Q_{min}}^{Q_{max}} Q\,[S(Q) - 1]\sin(Qr)dQ$$

## Neutron scattering experiments

Neutron diffraction and pair distribution function (PDF) measurements were performed on the NOMAD beamline at the Spallation Neutron Source (SNS), Oak Ridge National Laboratory. Approximately 0.15–0.2 g of h-[11]BN powders (AMBN treated at 600, 700, and 800 °C) were sealed in 3 mm thin-walled quartz capillaries. For each sample, four 24 min scans were acquired and summed to improve data statistics. Background from the empty capillary was subtracted, and the resulting spectra were normalized to a 6 mm vanadium standard to correct for detector efficiency. PDF data were obtained by Fourier transforming S(Q) to G(r) (or g(r)) with a Qmax of 50 Å$^{-1}$ for all samples.

Structural refinements were performed using TOPAS v6[100]. For the small-box neutron PDF analysis, g(r) data were modeled with a Lorentzian damping function [Exp($-$ r*Q$_{damp}$/2)] and a Pseudo-Voigt-like peak profile. An additional empirical term ($- \delta/r^2$) was included to account for low-r peak sharpening arising from correlated atomic motion.

For Rietveld refinement of the Bragg diffraction data, time-of-flight (TOF) patterns were converted to d-spacing using the relation TOF = ZERO + DIFC*d + DIFA*d$^2$, where ZERO is a constant, DIFC the diffractometer constant, and DIFA an empirical correction for peak shifts from sample displacement and absorption. ZERO and DIFC values were obtained from refinement of a NIST Si-640e standard and fixed, while DIFA was refined to account for sample displacement. The moderator-induced asymmetrical line profile was modeled using a modified Ikeda-Carpenter-David function[101,102], the peaks are then convoluted with a Pseudo-Voigt function to describe the specimen

induced peak broadening (microstrain and size).[103] Lorenz factor is corrected by multiplying $d^4$ [104].

## Semihydrogenation reaction of acetylene

Acetylene semihydrogenation experiments were done using an Altamira instruments 200 catalyst characterization flow reactor (AMI-200). Each acetylene semihydrigenation reaction was carried out with 75 mg of catalyst diluted with 75 mg of quartz and a reactant concentration of 0.5% $C_2H_2$/16% $H_2$/83.5% Ar with a flow rate of 50 ccm to give a gas hourly space velocity (GHSV) of 4000 $h^{-1}$. For reaction parameter screening, either the flow rate (25 ccm) or the catalyst loading (300 mg) varied, while all other conditions kept constant. The change of $C_2H_2$, $C_2H_4$, and $C_2H_6$ conversions and selectivities were calculated according to inlet and outflow rates measured during the reaction as follows:

$$C_2H_2\,conversion(\%) = ([C_2H_2]_{initial} - [C_2H_2]_{outlet})/[C_2H_2]_{initial} \times 100$$

$$C_2H_4\,selectivity(\%) = [C_2H_4]_{outlet}/([C_2H_2]_{initial} - [C_2H_2]_{outlet}) \times 100$$

$$C_2H_6\,selectivity(\%) = [C_2H_6]_{outlet}/([C_2H_2]_{initial} - [C_2H_2]_{outlet}) \times 100$$

Similar conditions were used for the acetylene/ethylene mixture reaction where 0.25% $C_2H_2$/10% $C_2H_4$/16% $H_2$/73.5% Ar was used at GHSV of 4000 $^{-1}$. However, since the ethylene outlet flow is already high due to the addition of 10% ethylene into the reaction, the $C_2H_4$ selectivity was calculated after correcting for the high outlet flow of $C_2H_4$ in the reaction as shown below:

$$C_2H_4\,outlet\,correction(ccm) = ([C_2H_4]_{outlet} - [C_2H_4]_{initial})$$

$$C_2H_4\,selectivity(\%) = ([C_2H_4]_{outlet} - [C_2H_4]_{initial})/ \\ ([C_2H_2]_{initial} - [C_2H_2]_{outlet}) \times 100$$

All other conversion and selectivity calculations for $C_2H_2$ and $C_2H_6$ remained the same. All semihydrogenation reactions were carried out at a total system pressure of 15 psi. Gas flow rates were calibrated and reported as actual volumetric flow rates (ccm) using an ADM volumetric flow meter, which accounts for deviations from standard temperature and pressure conditions. Additionally, all reactions were repeated at least three times, and the values reported in this manuscript represent the mean with the corresponding standard deviations.

**In-situ acetylene semihydrogenation diffuse reflectance infrared fourier transform spectroscopy (DRIFTS).** In-situ acetylene semihydrogenation DRIFTS was performed on a Thermo Fisher Nicolet iS50 FTIR spectrometer equipped with an MCT detector cooled by liquid nitrogen. To begin, the BN-700 sample was pretreated in a Harrick Scientific diffuse reflectance cell equipped with a temperature controller at 150 °C in 50 ccm He for 1 hour. After pretreatment, the background spectra were taken at 25 °C. Next, 1% $C_2H_2$/Ar at 25 ccm and 5% $H_2$/Ar at 25ccm (total flow: 50 ccm) was flowed through the catalyst for 10 min at 25 °C, followed by desorption with 50 ccm He for 10 min. Each spectrum was recorded in 1 min intervals during the adsorption and desorption process with 50 scans at a resolution of 0.4 $cm^{-1}$. The same procedure as noted above was done for 100 °C and 150 °C where each spectrum was taken without cooling back down to 25 °C.

## Gaseous $H_2$/$D_2$ exchange

The AMI-200 was utilized for the $H_2$/$D_2$ exchange experiments. In the experiment, the catalytic reactor was loaded with 75 mg of catalyst and subjected to a 50 ccm of 1:1 ratio of $H_2$/$D_2$ (24 % $H_2$/24% $D_2$/52% Ar) and

a heating rate of 10 °C $min^{-1}$. The reactor exit compounds ($H_2$, $D_2$, and HD) were quantified using online mass spectrometry with m/z values of 2, 4, and 3, respectively. The values reported in this manuscript represent the mean with the corresponding standard deviations.

## Computational methods

Spin-polarized density functional theory (DFT) calculations were carried out with the Vienna Ab initio Simulation Package (VASP)[105,106]. The generalized-gradient approximation (GGA) using the Perdew–Burke–Ernzerhof (PBE) functional was applied for exchange–correlation, with van der Waals interactions included via the Grimme DFT-D3 correction[107,108]. Electron–core interactions were treated using the projector-augmented wave (PAW) method, and a plane-wave cutoff energy of 400 eV was employed[109,110]. A 6×6×1 h-BN supercell was used to model defects and reactions, sampled only at the Γ-point. Transition states were identified using the climbing-image nudged elastic band and dimer methods, with a force convergence threshold of 0.05 eV $Å^{-1}$ [111,112].

## Data availability

The data that support the findings of this study have been included in the main text and Supplementary Information. The datasets generated during and/or analysed during the current study are available from the corresponding author on reasonable request. Source data are provided with this paper.

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

## Acknowledgements

The research was supported financially by the U.S. Department of Energy, Office of Science, Office of Basic Energy Sciences, Chemical Sciences, Geosciences, and Biosciences Division, Catalysis Science Program. A part of this work (solid-state NMR) was supported by the U.S. Department of Energy (DOE), Office of Science, Basic Energy Sciences, Materials Science and Engineering Division. Part of this work was conducted at the NOMAD beamlines at ORNL's Spallation Neutron Source, which was sponsored by the Scientific User Facilities Division, Office of Basic Sciences, U.S. Department of Energy. The research was performed at the Ames National Laboratory, which is operated for the U.S. DOE by Iowa State University under contract # DE-AC02-07CH11358. This research used resources of the Advanced Photon Source, a DOE User Facility, operated for the DOE Office of Science by the Argonne National Laboratory under contract no. DE-AC02-06CH11357. Mahadhi Hasan Khan, Renuka Mudiyanselage, and Qasim Adesope (UNT) are gratefully acknowledged for their assistance in XPS data acquisition. TW acknowledges Jacob Kinnun Core Facility Manager at the Institute for Advanced Materials & Manufacturing (UTK) for his support in TG-MS tests.

## Author contributions

Z.Y. conceived and designed the experiments. T.W. and K.M.S. led the paper writing, and all other authors contributed to the text. T.W. led material synthesis. K.M.S. led catalysis tests and F.P.-G. participated in measurements. M.L., S.Y, G.D. performed TEM and STEM tests. J.L. performed neutron scattering tests and analyses. A.S.I. performed high energy X-ray scattering tests and PDF analysis. Z.Z. and J.G. performed XAS tests and analysis. K.T. and S.A. performed $H_2$-DRIFTS tests and analysis. A.A.M. and J.K. performed XPS tests and analysis. H.S. and D.J. performed DFT calculation. T.K. performed NMR tests and analysis. S.D. supervised the project. T.W. an K.M.S. contributed equally to this work.

## Competing interests

The authors declare no competing interests.
