## [Transparent Peer Review file · Nature Communications]

Selective Semihydrogenation of Acetylene in Ethylene Using Defect-Rich Boron Nitride Catalyst from Flux Reconstruction

Corresponding Author: Professor Sheng Dai

Version 0:

Reviewer comments:

Reviewer #1

(Remarks to the Author)

The manuscript titled "elective Semihydrogenation of Acetylene in Ethylene Using Defect-Rich Boron Nitride Catalyst from Flux Reconstruction" by Wang et al presents the exploitation of defects in boron nitride to afford low temperature partial hydrogenation of acetylene.

The paper discusses a route to reproducibly preparing defect rich hexagonal boron nitride with particular activity toward partial hydrogenation. The paper, as is, requires revision by addressing the following comments.

1) Throughout the paper The authors discuss selectivity, with yield briefly shown inn Figure 5b and a. Neither CeO nor BN seem to be particularly active catalysis. Is 30% conversion of use? How might this be improved.

2) Page 2 line 63, What is the nature of the deviation from ordered BN? There are many types of defects that result in deviation only two values are presented.

3) Page 2 line 63, -505 kJ/mol and -907 kJ/mol are very high values and seem out of place from the numerous DFT studies on the binding of H₂ to defect structures in h-BN. Neither references 34 or 35 report these numbers for metal-free BN. The authors own paper in ref 35 reports a values of ~ -7.7 eV for Pt on a defect.

Other people report much lower values for the binding energy of H₂ on a vacancy.

For instance:

Your ref 38:

VB: -4.95 eV

VN: -1.43 eV

Shevlin, S.A. and Z.X. Guo, Hydrogen sorption in defective hexagonal BN sheets and BN nanotubes. Physical Review B, 2007. 76(2): p. 024104.

VB: -5.18 eV

VN: -1.55 eV

4) Page 2 line 68, Of the references 34, 36, 37, 38, 39 ,40, and 41, only one paper (38) presents hydrogen activation and it is not by the FLP mechanism. Reference 38 should be in the references line 65. perhaps omitting ref 34 as it involves a metal and is not relevant to the discussion.

Summary of references:

34 - Pt on BN

36 - DFT study

37 - fabrication of defect no hydrogen absorption discussion

38 - implementation of hydrogenation over in situ defects

39 - hydrogen storage not activation

40 - adsorption on defect-free BN

41 - hydrogen adsorption on porous BN - defects not discussed nor is FLP

5) Page 2 line 68 A single atom vacancy would not behave as an FLP.

6) Page 3 Figure 1d A Williamson-Hall analysis of the restructured sample would give a measure of the strain and perhaps more information on the concentration of defects in the material.

7) Page 4 line 103 The borax-urea method is the most common way of producing h-BN commercially. Boric acid is produced from borax through acid treatment.

8) Page 5 line 161 If the goal was to produce oxide and impurity free BN, the NaNH_2 washing procedure would potentially introduce oxides and halides onto the surface. Solid state NMR is relatively insensitive. An oxygen (an other heteroatoms) incorporation level of a few percent may not be observable. It is stated in page 6 line 182 the most (but not all?) of the O was removed during NaNH_2 processing.

9) Page 5 line 168 - can the N-H bonds be unambiguously assigned to in sheet defects? How much do edge atoms contribute to this signal.

10) Page 7 line 217. The difference in residues of the neutron PDF analysis are small. Especially given that the defect free structure has the lowest Rwp. This approach seems to neglect the possibility of multiple defect-types being present. It also does not identify the defect most likely to be active for hydrogenation.

11) Page 9 line 251 "Previous works demonstrated the critical roles of the open B and N sites to activate hydrogen molecules via the FLP manner and adsorb the substrates." Which previous works?

12) Page 10 line 277. If the active sites bind the hydrogen so strongly (as the TPD data suggests) how is any scrambling observed?

13) Page 12 line 347. Hydrogenation over BN has been observed before and modeled. Previous studies found that hydrogenation occurred on olefinic carbons but not aromatic. The most important aspect of this report is the fact that hydrogenation is trapped at ethylene formation. Why? What structure functionality favors this?

14) Supplementary: NaNH_2 is extremely corrosive, very few material can withstand molten NaNH_2 . Was any nickel incorporated in the structure (ICP-MS analysis?). This could lead to the incorporation of impurities in the BN structure.

15) Supplementary: It should be noted that NaNH_2 reacts vigorously with water. Care must be taken during the washing step.

16) Supplementary: What was the pressure of the semihydrogenation reactor? Was the flow 50 ccm, pressure corrected, or 50 sccm, uncorrected

Reviewer #2

(Remarks to the Author)

The authors reported an impressive work on the fabrication of BN-700 catalyst by the from flux reconstruction method, which exhibited good reactivity in the conversion of semihydrogenation of acetylene in ethylene. The catalyst after the treatment with NaNH_2 was identified with abundant open B, N sites and defects through different characterization technologies, the structure was also simulated to provide more detail information regarding the defects types and stacking models. The catalytic reactivity in activating H_2 was confirmed by the H-D exchange assay, suggesting the high potential in hydrogenation reactions. Appealing, the catalyst was able to hydrogenate a low concentration of acetylene in ethylene, which was an important industrial application, and the reactivity surpass the metal oxide catalyst CeO_2 . The catalyst performances was also detailed investigated. The paper was well structure and the data was also comprehensive and convincing. Overall, the paper can be accepted for publication after the minor points are addressed.

1, As for the metal-free catalytic system for acetylene semihydrogenation, the authors could systematically study the literatures and provided the most relevant literatures in this area, to better show the communities the important advance on this specific topic (SI is also fine).

2, As for the metal-free catalyst, people may care about the existence of trace amount of metals, which sometimes may serve as the catalytic sites for hydrogenation. To verify that, the authors are suggested to carry out the ICP analysis of the relevant catalysts to exclude this.

3, The side product ethane was also noticed in the reaction, is it possible to get a higher selectivity by optimizing the reaction conditions? Besides, what's the reason or principle for the composition of the mixed gas? Especially the hydrogenation ratio in the mixed gas.

4, The releasing of the ethylene from catalytic sites is also important to the high selectivity. Although the authors provided calculation results about the low tendency of ethylene adsorption on catalyst surface. Is it possible to investigate the adsorption behaviors of acetylene and ethylene on catalyst surface and compare that?

5, A schematic catalytic cycle of the reaction pathway could be provided to better show the reaction mechanism.

Reviewer #3

(Remarks to the Author)

Ethylene purification remains a long-standing challenge in various industrial processes where ethylene serves as a fundamental building block. Conventional catalysts for selective acetylene hydrogenation primarily rely on noble and transition metal active sites, which are constrained by high costs and poor thermal stability. In this work, Dai et al. introduce a novel metal-free catalyst for the highly selective semihydrogenation of acetylene in ethylene streams. The defect-rich boron nitride (BN) catalyst was synthesized via flux reconstruction, and its structural properties, including defect types and stacking modes, were comprehensively characterized. BN-700 exhibited remarkable H₂ dissociation activity in both liquid and solid phases, leading to highly selective acetylene semihydrogenation. The low energy barrier for H₂ activation, coupled with accelerated ethylene desorption to prevent over-hydrogenation, was validated through isotope studies and theoretical simulations. Given the well-structured presentation and significant scientific contribution, I recommend acceptance of this manuscript after minor revisions.

1. The flux reconstruction process for converting AMBN into BN nanoparticles is impressive, particularly in achieving the removal of O and C impurities. However, it remains unclear where these impurities go. A TG-MS (thermogravimetric-mass spectrometry) experiment is recommended to analyze the gaseous byproducts and provide insights into the fate of O and C impurities.

2. While the manuscript maintains consistent terminology, some technical terms require clearer definitions for broader readability. For instance, the terms "BN₂ vacancy" and "BN₃ vacancy" should be explicitly defined in the introduction or methods section to assist non-expert readers.

3. The statement "The AMBN precursor displayed the presence of tetracoordinated BN_xO_{4-x} (x = 0-4) species, together with the regular tricoordinate BN₃ bonds (Figure 2a)" introduces the concept of tetracoordinated BN_xO_{4-x} species, which may be unfamiliar to some readers. While the BN₃ bonds are well understood, an illustration or reference supporting the tetracoordinated BN_xO_{4-x} structure would enhance clarity.

4. The DFT calculations effectively support the experimental findings. However, it would be beneficial to compare the computed H₂ activation energy with previously reported catalysts to contextualize the performance of BN-700 relative to existing materials.

5. Some recent progress in selective semihydrogenation should be cited, including J. Am. Chem. Soc. 2021, 143, 4483, Angew. Chem. Int. Ed. 2024,63, e202410394, which should enhance the manuscript's readability.

Version 1:

Reviewer comments:

Reviewer #1

(Remarks to the Author)

The authors have not adequately addressed the issues brought up in the previous review. In particular, this manuscript inappropriately cites previous olefin hydrogenation work over defects in boron nitride (reference 46). This is especially important as key points in this manuscript are not supported by theoretical and experimental results in the prior work.

1) A conversion of at most 40% will not reduce a stream containing 0.25-2% down to 0.00005%. That would require a conversion of 99.98%. It seems the path forward will be challenging as efforts to increase conversion led to subsequent reduction in selectivity.

Titanium doping is presented in the reviewer response but is not in the manuscript materials. Is this supplied as extra information for the reviewers? It seems to introduce more questions than answers. Specifically how the authors could be sure titanium is in the structure of the BN and not converted to the oxide during removal of the NaNH₂.

2,3)

As the authors point out the conversion from eV to kJ requires proper conversion. Attributing the discrepancy between their values of -505 kJ/mol and -907 kJ/mol and literature values neglects the fact that these values convert to -5.23eV and -9.40eV; far stronger binding than most of the values they presented in table S2.

Minor discrepancies are to be expected with different researchers using slightly different parameters. However a factor of more than 2 suggests an error unless the authors have some overwhelming evidence that previous work was performed in an incomplete manner.

In the manuscript "Theoretical simulations indicate that BN scaffolds with vacancies deviating from the ideal B₃N₃ structure exhibit markedly enhanced H₂ adsorption, with binding energies as low as -5.18 eV," Ignores the 3rd entry in the table - Reference S20 which presents values for H₂ adsorption on nanotubes that are much higher than previous work. Shevlin et al reports Eads nanotube VB: -4.10; VN: -0.74

It is also strange that the authors choose to report H₂ Gibbs free energy of Hydrogenation from reference S23 when there are numerous studies that report the energy of adsorption.

It should also be noted that very negative binding energies will disfavor hydrogenation and favor surface hydrogenation. Binding energies should be in the range of Pt, Pd and Ni, which are around -1.6 to -0.3 eV. In this case high binding energy is not the best, one wants a moderate binding energy.

4)

Page 3 - reference S24 states that single atom defects do not activate by a FLP mechanism.

Again the authors do not present the reference to previous work in a clear manner.
references 42-46

42- is not BN but the precursor with excess B or N

43- reports larger than single atom defects

44 - carbon scaffold with B and N functionality

45 - BCN not BN

46 - hydrogenation of olefins over defects - directly related to the presented work

Ref 46 specifically states that the mechanism is not an FLP mechanism with DFT models supporting a Horiuti-Polanyi like mechanism. This 2016 paper is corroborated by ref S24.

This work needs to be discussed in a separate sentence. Something like:

Although complete hydrogenation of olefins over defects generated in situ was reported by Nash et al[46], we achieved semi-hydrogenation through implementation of an flux reconstruction process to ...[insert insight]

12) The point of this comment was that the authors' models indicate strong hydrogen binding to active sites. This is counter to the H₂/D₂ scrambling observed and more in line with previous theoretical calculations showing binding energies from -.5 to -4.95. H₂/D₂ scrambling would be more plausible if binding energies were in the -2.3 to -.3 eV range.

13) The explanation is counter the work presented in reference 46. In reference 46, propylene and cyclohexene were fully hydrogenated under fairly mild conditions. While compounds such as diphenethylene were only hydrogenated at the olefinic bond. The olefinic bond on ethene absolutely interacts with vacancies in the BN structure in a eta-2 interaction. As per ref 46 ethene's binding energy on defects is on par with propene:

ethene VB: -3.71, VN -1.90

Propene VB: -3.69, VN -1.76

DFT models even show electron density sharing between the olefinic bond and the defect.

In the DRIFTS explanation how would acetylene hydrogen bond to a surface when all the atoms in acetylene are expected to have partial charges of 0?

It is unfortunate that the key vibrational energy of the C=C stretch (1623 cm⁻¹) in ethene is mostly lost in the absorption data. Although there does seem to be a small peak shifted to lower energy (as would be expected in eta-2 binding)

A better way to assess this binding is to saturate a catalyst sample with ethene and perform transmission FTIR in a KBR pellet mixed with a little powder. This worked well for CO₂ adsorbed on a BN surface.

Chagoya, K.L., D.J. Nash, T. Jiang, D. Le, S. Alayoglu, K.B. Idrees, X. Zhang, O.K. Farha, J.K. Harper, T.S. Rahman, and R.G. Blair, Mechanically Enhanced Catalytic Reduction of Carbon Dioxide over Defect Hexagonal Boron Nitride. ACS Sustainable Chem. Eng., 2021. 9(6): p. 2447-2455.

Additional Comments:

A) Page 3 line 87: 14 nm crystallite sizes are a quite small and not indicative of a highly crystalline material. Especially when the region where amorphous halo for BN shows up in XRD is not in the plot.

B) Page 5 line 153 - BN is not classically aromatic as pi electron density is localized around the nitrogen atoms.

C) Supplemental Figure 3:A Williamson-Hall plot is produced from the integral breadths of each peak not the FWHM.

D) The figures in the revised manuscript are out of order.

E) In figure 5 ionothermal is misspelled.

Reviewer #2

(Remarks to the Author)

The authors have made major revisions to their previous submission, which well addressed the questions raised by the reviewers and made the research work more convincing. I think it can be accepted for publication now.

Reviewer #3

(Remarks to the Author)

This manuscript presents a rigorous and well-supported study on selective semihydrogenation of acetylene on defect-rich boron nitride catalysts, offering valuable insights. The revised paper looks very nice and the conclusions are well-justified. I recommend publication without further revision.

Version 2:

Reviewer comments:

Reviewer #1

(Remarks to the Author)

The revised paper is suitable for publication. Although, I am not fully convinced by the proposed mechanism the authors do present data that suggests the rate of hydrogenation of the first pi bond in acetylene is much faster than the subsequent pi bond.

A few comments on the authors response follow:

0) In reference to comment "Ref 46 specifically states that the mechanism is not an FLP mechanism with DFT models supporting a Horiuti–Polanyi like mechanism. This 2016 paper is corroborated by ref S24."

Correction reference S21

"We find that H₂ activation by FLP is the most favorable type of bonding only if the defect size is large enough to accommodate steric effects between H atoms"

1) The feature of this mechanochemical process is the localized extreme high temperature and mild overall system temperature.

This is a very outdated reference and view of the mechanism of mechanochemical reactions. Much work has shown that the hot spot mechanism is highly unlikely. For example:

James, S.L., C.J. Adams, C. Bolm, D. Braga, P. Collier, T. Friščić, F. Grepioni, K.D.M. Harris, G. Hyett, W. Jones, A. Krebs, J. Mack, L. Maini, A.G. Orpen, I.P. Parkin, W.C. Shearouse, J.W. Steed, and D.C. Waddell, Mechanochemistry: opportunities for new and cleaner synthesis. *Chemical Society Reviews*, 2012. 41(1): p. 413-447.

<https://doi.org/10.1039/C1CS15171A>

"It seems unlikely that hot spots and magma-plasma sites are the primary sites of reactivity in molecular organic and metal–organic mechanochemical reactions. If they were, extensive decomposition would be expected. That such decomposition is not seen suggests that these phenomena may be too brief and/or too localized to be the primary reactive sites for molecular organic reactions. "

This assertion is corroborated by ref 46 where high temperatures would have resulted in severe coking - which was not evident.

A more likely explanation of the difference is that the manuscript presents work performed in a plug flow reactor whereas Nash et al reported results from a batch reactor with a large amount of catalyst (5g). In such a case complete hydrogenation would be in line with the author's observation that increases catalyst loading resulted in increased ethane yield.

2) Although there is work showing hydrogen bonding between various C-H groups with oxygen-containing species. It is not necessarily established with NH species especially given the lower bond energy that would be expected.

Responses to Reviewers' Comments

Reviewer #1 (Remarks to the Author):

The manuscript titled "Selective Semihydrogenation of Acetylene in Ethylene Using Defect-Rich Boron Nitride Catalyst from Flux Reconstruction" by Wang et al presents the exploitation of defects in boron nitride to afford low temperature partial hydrogenation of acetylene. The paper discusses a route to reproducibly preparing defect rich hexagonal boron nitride with particular activity toward partial hydrogenation. The paper, as is, requires revision by addressing the following comments.

1) Throughout the paper the authors discuss selectivity, with yield briefly shown in Figure 5b and a. Neither CeO₂ nor BN seem to be particularly active catalysis. Is 30% conversion of use? How might this be improved.

Response: We thank the reviewer for the comments regarding the conversion. The optimized reaction conditions (75 mg catalyst; gas mixture: 0.5% C₂H₂ / 16% H₂ / 83.5% Ar; flow rate: 50 ccm; gas hourly space velocity (GHSV): 4000 h⁻¹; temperature: 150 °C), leading to around 30% acetylene conversion, were determined after comprehensive screening of reaction parameters, including flow rate, catalyst loading, and reaction temperature. The primary objective was to achieve high acetylene conversion while maintaining ethene selectivity above 95%. As outlined in the Introduction part, the goal of acetylene hydrogenation is to remove trace acetylene impurities from ethene streams to obtain high-purity ethene. In industrial applications, naphtha cracking typically produces ethylene streams containing 0.25–2% acetylene impurities (*Materials Today Catalysis* 2023, **3**, 100029; *ACS Catalysis* 2019, **9**, 8592). Reducing acetylene concentrations from ~2% to below 5 ppm is essential yet challenging for ensuring the purity required in downstream processes such as the production of ethylene oxide and polymers. Therefore, as long as the catalyst exhibits sufficient activity for acetylene hydrogenation and maintains high ethene selectivity, moderate conversion levels are acceptable, as purification can be achieved through repeated hydrogenation steps. As demonstrated in our catalysis results, the as-synthesized BN700 catalyst retained its activity and selectivity under conditions mimicking an ethene-rich stream (0.25% C₂H₂/10% C₂H₄/16% H₂/73.5% Ar, 150 °C, GHSV 4000 h⁻¹).

As suggested by the reviewer, the following efforts have been made to clarify the reaction condition screening and further improve the catalytic activity:

- (1) Detailed results of the reaction condition screening have been included in the **main manuscript (Page 9)** and the **Supporting Information (Page S17 and Supplementary Fig. 25)**.

Manuscript (Page 9)

To enhance catalyst performance, variations in catalyst loading and flow rate were systematically investigated to identify optimal conditions for improved activity and ethylene selectivity. Reducing the flow rate to 25 ccm led to an increase in acetylene conversion from 30% to 40%; however, ethylene selectivity

dropped from 98% to 81% (Supplementary Fig. 25A). In a separate experiment, increasing the catalyst loading from 75 mg to 300 mg improved conversion from 30% to 46%, but similarly reduced ethylene selectivity to 76% (Supplementary Fig. 25B). For practical acetylene semihydrogenation, the objective is to remove trace acetylene from ethylene streams while preserving high ethylene purity. It is crucial to maintain ethene selectivity above 95%. Therefore, the optimized reaction conditions selected for further studies were: 75 mg catalyst; gas composition: 0.5% C₂H₂/16% H₂/83.5% Ar; flow rate: 50 ccm; gas hourly space velocity (GHSV): 4000 h⁻¹; and reaction temperature: 150 °C.

Supporting Information (Page S17 and Supplementary Fig. 25)

Acetylene semihydrogenation experiments were done using an Altamira instruments 200 catalyst characterization flow reactor (AMI-200). Each acetylene semihydrogenation reaction was carried out with 75 mg of catalyst diluted with 75 mg of quartz and a reactant concentration of 0.5% C₂H₂/16% H₂/83.5% Ar with a flow rate of 50 ccm to give a gas hourly space velocity (GHSV) of 4000 h⁻¹. For reaction parameter screening, either the flow rate (25 ccm) or the catalyst loading (300 mg) was varied, while all other conditions were kept constant.

Supplementary Fig. 25. Screening of reaction parameters for acetylene hydrogenation catalyzed by BN-700. (A) 75 mg of BN-700 catalyst using a 25 ccm flow of 0.5% C₂H₂ / 16% H₂ / 83.5% Ar at 150 °C. (B) 300 mg of BN-700 catalyst using a 50 ccm flow of 0.5% C₂H₂ / 16% H₂ / 83.5% Ar at 150 °C.

(2) Preliminary studies were conducted to further enhance the catalytic performance of BN via Ti doping.

Theoretical simulations indicated that Ti doping in BN could significantly improve H₂ adsorption energy from approximately -0.1 eV to -0.7 eV (*Applied Physics Letters*, 2006, **89**, 153104). To investigate this, a Ti-doped BN (Ti-BN) catalyst was synthesized via a modified flux reconstruction method (**Figures R1A**). Preliminary acetylene hydrogenation tests showed that doping 10% of the Ti sites in BN led to a 12% increase in acetylene conversion compared to the BN-700 catalyst under identical conditions. However, this improvement came with a 6% decrease in ethylene selectivity (**Figure R1B**). These findings suggest that heteroatom doping is a promising strategy to fine-tune the catalytic activity and selectivity of BN. Nonetheless, further development is needed to create metal-free BN catalysts with performance comparable to that of noble- and transition-metal-based systems. Future efforts will focus on exploring alternative dopants (e.g., P and F), optimizing dopant concentrations, tuning porosity, and refining catalyst morphology.

Figure R1 (A) XRD pattern of Ti-BN synthesized through a flux method. (B) C₂H₂ hydrogenation over 10% Ti-doped BN catalyst using 75 mg catalyst with 50 ccm of 0.5% C₂H₂/16% H₂/83.5% Ar mixture over a temperature range of 100 to 150 °C.

2) Page 2 line 63, What is the nature of the deviation from ordered BN? There are many types of defects that result in deviation only two values are presented.

Response: We thank the reviewer for the insightful comments on defects in BN and the reported H₂ adsorption energies in Comments 2 and 3. Our detailed responses to these comments have been consolidated in the response to Comment 3.

3) Page 2 line 63, -505 kJ/mol and -907 kJ/mol are very high values and seem out of place from the numerous DFT studies on the binding of H₂ to defect structures in h-BN. Neither references 34 or 35 report these numbers for metal-free BN. The authors own paper in ref 35 reports a values of ~-7.7 eV for Pt on a defect. Other people report much lower values for the binding energy of H₂ on a vacancy.

For instance: Your ref 38: VB: -4.95 eV, VN: -1.43 eV

Shevlin, S.A. and Z.X. Guo, Hydrogen sorption in defective hexagonal BN sheets and BN nanotubes. Physical Review B, 2007. 76(2): p. 024104.

VB: -5.18 eV, VN: -1.55 eV

Response: The apparent discrepancies in the reported values primarily arise from the use of different units, either eV or kJ/mol (with the conversion factor: kJ/mol = eV × 96.485). Additionally, most H₂ adsorption energy values for BN have been obtained through theoretical simulations, and variations in computational methods can lead to differences even for the same defect structures. To clarify the rationale and provide context, we have included a table summarizing the reported H₂ adsorption energies for BN with various defect types. The relevant descriptions and references have also been revised and reorganized accordingly.

Manuscript (Page 2)

For example, in perfect sp²-hybridized B–N hexagonal units, the binding energies of H₂ were calculated to be –0.085 eV and –0.1 eV at the B and N sites, respectively.³⁶ Theoretical simulations indicate that BN scaffolds with vacancies deviating from the ideal B₃N₃ structure exhibit markedly enhanced H₂ adsorption, with binding energies as low as –5.18 eV, depending on the vacancy size and atomic configuration (Table S1). These defects, composed of unsaturated B and N sites, can activate and dissociate H₂ molecules through the formation of N–H and B–H bonds, facilitating subsequent hydrogenation reactions.^{37–40} In addition, the physical, chemical, and electronic properties of h-BN can be micro-engineered through tuning crystallinity, exfoliation, defect creation, dopants addition, and heteroatom incorporation.⁴¹

Supporting Information (Page S19, Supplementary Table 1)

Supplementary Table 1 Summary of H₂ adsorption energies on various boron nitride materials based on theoretical simulations.

Sample	Defect types	Energy type	Value from simulation	Reference
Boron Nitride nanotubes	B sites and N sites	Adsorption energy	-0.085 and -0.1 eV	S18
Boron Nitride nanotubes	B sites and N sites	Adsorption energy	-0.027 and -0.037 eV	S19
Boron Nitride Nanotubes	V _N , V _B	Adsorption energy	-5.24 and -9.40 eV	S20
Boron Nitride Nanotubes	V _N , V _B	Adsorption energy	-1.55 and -5.18 eV	S21
hexagonal boron nitride ^[a]	BN, BN ₂ , BN ₃ , B ₃ N ₃ , B ₇ N ₆ , B ₇ N ₁₂	Dissociative H ₂ adsorption energy	-5.1 to 0.5 eV	S22
hexagonal boron nitride ^[b]	2V(1B1N), 3V(2B1N), 3V(1B2N), 4V (3B1N), 6V(3B3N)	Gibbs free energy of hydrogenation	-4.0 to -0.5 eV	S23
Carbon-doped hexagonal boron nitride ^[c]	C _B (N,N,N), C _B (C,N,N), C _B (C,C,N), C _B (C,C,C), C _N (B,B,B), C _N (C,B,B), C _N (C,C,B), C _N (C,C,C)	Adsorption energy	-0.084 to -0.063 eV	S24
Li ⁺ doped boron nitride nanotubes	NA	Adsorption energy	-0.33 eV	S25

lithium decorated 3D hybrid Boron-Nitride-Carbon frameworks	NA	Adsorption energy	-0.24 eV	S26
Ti-doped boron nitride	NA	Adsorption energy	-0.70 eV	S27
[a] Structure of boron nitride with defects shown in Supplementary Fig. 10.				
[b] Structure of boron nitride with defects.				
				
[c] Structure of carbon-doped boron nitride.				

4) Page 2 line 68, of the references 34, 36, 37, 38, 39, 40, and 41, only one paper (38) presents hydrogen activation and it is not by the FLP mechanism. Reference 38 should be in the references line 65. perhaps omitting ref 34 as it involves a metal and is not relevant to the discussion. Summary of references: 34 - Pt on BN, 36 - DFT study, 37 - fabrication of defect no hydrogen absorption discussion, 38 - implementation of hydrogenation over in situ defects, 39 - hydrogen storage not activation, 40 - adsorption on defect-free BN, 41 - hydrogen adsorption on porous BN - defects not discussed nor is FLP.

5) Page 2 line 68 A single atom vacancy would not behave as an FLP.

Response: We thank the reviewer for the insightful comments regarding the citation of FLP systems and the nature of active sites in FLP, as mentioned in **Comments 4) and 5)**. In response, and to provide a clearer overview of previous studies on FLP systems, we have included a table (**Supplementary Table 2**) summarizing key achievements in FLP-catalyzed reactions involving small gas molecules. The table covers systems based on boron nitride, heteroatom-doped carbon scaffolds, metal oxides, semi-heterogeneous catalysts, and selected homogeneous catalysts. The main text has been reorganized accordingly to incorporate this clarification.

Manuscript (Page 3)

Notably, the high structural tunability of h-BN enables the controlled formation of unsaturated boron and nitrogen sites in close proximity, which can act as strong acid–base centers to activate small gas molecules via frustrated Lewis pair (FLP)-like behavior (Tables S2). Such mechanisms have been demonstrated in previous studies using h-BN derived from ionothermal synthesis, as well as B- and N-doped carbon materials.⁴²⁻⁴⁶ Furthermore, the activation of small gas molecules via coexisting acid and base sites has also been reported in FLP catalysts based on heteroatom-doped carbon scaffolds,^{47,48} CeO₂,⁴⁹ and InOH,⁵⁰ as well as in semi-heterogeneous^{51,52} and homogeneous systems.^{53,54}

Supporting Information (Page S20, Supplementary Table 2)

Supplementary Table 2 Selected examples of the previously reported heterogenous and homogeneous catalytic FLP systems.

FLP catalyst	Structure	Acid sites	Base sites	Reaction	Ref.
Heterogeneous s		B sites	N sites	Hydrogenation of alkenes and alkynes	S28
Heterogeneous s		B sites	N sites	Hydrogenation of alkenes and alkynes	S29
Heterogeneous s		B sites	N sites	Olefin hydrogenation via mechanochemistry	S30
Heterogeneous s		B sites	N sites	Selective Hydrogenation	S31
Heterogeneous s		B sites	N sites	Electrocatalytic nitrogen reduction	S32
Heterogeneous s		C sites	N sites	Dehydrogenation	S33
Heterogeneous s		C sites	Heteroatoms	Alkene hydrogenation	S34

Heterogeneous		surface defects (adjacent surface Ce ³⁺)	surface lattice oxygen	Hydrogenation of styrene	S35
Heterogeneous		In(III) sites	InOH sites	Hydrogenation of CO ₂	S36
Semi-heterogeneous		B(C ₆ F ₅) ₃	Oxygen atoms on the surface of the α-cyclodextrin	Hydrogenation, Deoxygenation	S37
Semi-heterogeneous		B(C ₆ F ₅) ₃	Pyridyl sites	Hydrogenation of ketones	S38
Homogeneous		B(C ₆ F ₅) ₃	electron-rich B atoms	Hydrogenation	S39
Homogeneous		2,2,6,6-tetramethylpiperidine (TMP)	B in H-B(C ₆ F ₅) ₃	Hydrogenation of CO ₂ to CH ₃ OH	S40

6) Page 3 Figure 1d A Williamson-Hall analysis of the restructured sample would give a measure of the strain and perhaps more information on the concentration of defects in the material.

Response: Thank you for your insightful comment. In accordance with the reviewer's suggestion, a Williamson-Hall analysis was conducted on the XRD pattern of BN700 to evaluate crystallite size, microstrain, and dislocation density. The relevant discussion and detailed calculation methods have been added to the manuscript and supporting information.

Manuscript (Page 4)

The average crystallite size (D), microstrain (ϵ), and dislocation density (δ) were determined from the XRD pattern of BN700 using the uniform deformation model (UDM),⁷⁰ based on the Williamson-Hall (W-H) and Williamson-Smallman (W-S) methods, after correcting for instrumental broadening with a silicon standard. This analysis assumes uniform strain across all crystallographic directions. The W-H plot and correction details are shown in Supplementary Fig. 3. The crystallite size was estimated to be 14.6 nm. The negative strain value ($\epsilon = -0.0095$) indicates compressive strain, likely due to lattice defects or dislocations, as further supported by the high dislocation density ($\delta = 4.7 \times 10^{15} \text{ m}^{-2}$).

Supporting Information (Page S2 and Supplementary Fig. 3)

Supplementary Fig. 3. Silicon standard XRD data (a), along with the calculated FWHM values for correcting instrumental broadening (b), and the Williamson-Hall plot of BN-700 after instrumental broadening correction (c).

Powder XRD (PXRD) was performed on a PANalytical Empyrean diffractometer, operated at 45 kV and 40 mA (scanning step: 0.02 ° per step). The diffraction patterns were collected in the range of 20-80°. $\lambda = 0.1540598 \text{ nm}$. The Williamson-Hall formula ($\beta_{\text{corrected}} \cos(\theta) = 4\epsilon \sin(\theta) + K\lambda/D$) was utilized to determine the average crystallite size (D) and micro-strain (ϵ) of BN-700 after correcting for instrumental broadening. Instrumental broadening was corrected with a Silicon powder standard where the full width half maximum (FWHM = β) of each peak was utilized for correction according to the following formula: $\beta_{\text{corrected}} = \sqrt{(\beta_{\text{BN-700}})^2 - (\beta_{\text{Silicon Standard}})^2}$. The uniform deformation model (UDM) was used to determine the dislocation density (δ) according to the Williamson-Smallman formula ($\delta = 1/D^2$) where it is assumed that that strain is uniformly distributed across all crystallographic directions.

7) Page 4 line 103 The borax-urea method is the most common way of producing h-BN commercially. Boric acid is produced from borax through acid treatment.

Response: We thank the reviewer for this clarification. The context was revised accordingly and related reference of h-BN synthesis using borax and urea was cited.

Manuscript (Page 3)

Pyrolysis treatment of organic raw materials containing B and N (e.g., urea, borax, boric acid, and melamine) could afford h-BN with large scale but relatively low crystallinity or amorphous nature (denoted as AMBN).⁵⁶⁻⁵⁸

8) Page 5 line 161 If the goal was to produce oxide and impurity free BN, the NaNH_2 washing procedure would potentially introduce oxides and halides onto the surface. Solid state NMR is relatively insensitive. An oxygen (an other heteroatoms) incorporation level of a few percent may not be observable. It is stated in page 6 line 182 the most (but not all?) of the O was removed during NaNH_2 processing.

Response: We thank the reviewer for the comment. X-ray photoelectron spectroscopy (XPS) was conducted to assess changes in surface oxygen content resulting from the flux reconstruction process.

Manuscript (Page 6)

The surface oxygen content decreased from 14.9 at.% in AMBN to 3.6 at.% in BN700, as determined by X-ray photoelectron spectroscopy (XPS, Supplementary Fig. 8).

9) Page 5 line 168 - can the N-H bonds be unambiguously assigned to in-sheet defects? How much do edge atoms contribute to this signal.

Response: We thank the reviewer for the valuable comment regarding the peak assignment in the solid-state ^{15}N NMR spectrum. We agree that the peak at -308 ppm may originate from both in-sheet defects and N-H terminated edge atoms. To clarify, due to the low natural abundance of ^{15}N ($\sim 0.4\%$), ^{15}N -labeled AMBN was synthesized using ^{15}N -labeled urea as the nitrogen source. However, for flux synthesis, regular (non-labeled) NaNH_2 was used as the reaction medium/catalyst. This means that all observable ^{15}N NMR signals in AMBN and BN700 originate solely from nitrogen atoms originally derived from the labeled urea precursor. Based on this, we conclude the following: (1) The signal at -308 ppm is attributed to $\text{NH}(\text{BN}_2)_2$ units at nitrogen-terminated zigzag edges. This signal is absent in AMBN due to its low concentration, below the detection limit. (2) If NaNH_2 contributes to nitrogen-terminated zigzag edges, these sites are not ^{15}N -labeled and thus do not appear in the ^{15}N NMR spectrum under our experimental conditions. (3) Therefore, the -308 ppm peak is assigned to edge defects where N-terminals, generated by C-N bond cleavage, are subsequently hydrogenated.

10) Page 7 line 217. The difference in residues of the neutron PDF analysis are small. Especially given that the defect free structure has the lowest R_{wp} . This approach seems to neglect the possibility of multiple defect-types being present. It also does not identify the defect most likely to be active for hydrogenation.

Response: We thank the reviewer for the valuable comment regarding the neutron PDF analysis. We would like to clarify that the R_{wp} value for the defect-free h-BN structure (9.1%) is close to that of the BN_2 -type vacancy model (9.08%) under the assumption of a perfectly planar structure without off-plane atomic displacements. However, when off-plane relaxation unique to defective structures is allowed in the BN_2 model, the R_{wp} significantly decreases to 5.7% (**Figure 3b**). This relaxation is absent in ideal h-BN, which consists of perfectly stacked 2D layers.

We agree with the reviewer that real h-BN catalysts likely contain a variety of defect types. However, neutron PDF analysis alone cannot resolve or quantify multiple coexisting defects due to its ensemble-averaged nature and the small free energy differences between defect configurations. While high-resolution microscopy (DOI: 10.1038/ncomms15291) can reveal individual defects, this technique requires ultrathin samples, which was not feasible with the relatively thick h-BN synthesized via flux reconstruction in our study. Our neutron PDF analysis (**Figures S10–S17**) was intended to compare structural models developed in our previous work (DOI: 10.1021/jacs.2c00343), and to identify the most likely dominant defect type (BN₂ in this case). We acknowledge that neutron PDF does not exclude the presence of other defect types. This was clarified in the manuscript.

Manuscript (Page 7)

It should be noted that h-BN catalysts may contain a variety of defect types. However, neutron PDF analysis cannot resolve or quantify multiple coexisting defects due to its ensemble-averaged nature and the small free energy differences between various defect configurations.

11) Page 9 line 251 "Previous works demonstrated the critical roles of the open B and N sites to activate hydrogen molecules via the FLP manner and adsorb the substrates." Which previous works?

Response: We thank the reviewer for the valuable comment. Relevant references on catalytic procedures involving BN or B- and N-doped catalysts in hydrogenation reactions have been cited.

Manuscript (Page 8)

Previous works demonstrated the critical roles of the open B and N sites to activate hydrogen molecules via the FLP manner and adsorb the substrates.^{42,44-46,68}

12) Page 10 line 277. If the active sites bind the hydrogen so strongly (as the TPD data suggests) how is any scrambling observed?

Response: We thank the reviewer for the valuable comment. The TPD data (**Figure 4a**) were obtained using CO₂ (acidic probe) and NH₃ (basic probe) to characterize the presence of basic and acidic sites respectively in the h-BN catalysts. These polar probe molecules interact strongly with the defect sites, revealing their acid–base character. In contrast, H₂ is a non-polar and relatively inert molecule with much weaker adsorption affinity. This weaker interaction allows for reversible adsorption and activation of H₂/D₂ on defect sites, enabling scrambling to occur. This is evidenced by the observed H₂/D₂ exchange in cyclohexane at 100 °C (**Figure 4c**) and in the gas phase at 250 °C (**Figures 4d and 4e**). Thus, while the defect sites can strongly bind polar molecules, they interact weaker with H₂, which facilitates catalytic exchange rather than inhibition due to over-binding.

13) Page 12 line 347. Hydrogenation over BN has been observed before and modeled. Previous studies found that hydrogenation occurred on olefinic carbons but not aromatic. The most important aspect of this report is the fact that hydrogenation is trapped at ethylene formation. Why? What structure functionality favors this?

Response: We thank the reviewer for raising this insightful point. This question has been addressed through two complementary approaches: theoretical simulations and additional DRIFTS experiments under acetylene/H₂ flow.

(1) Clarification of simulation discussion

The key distinction in the present case of acetylene semihydrogenation lies in the weak adsorption strength of ethylene on the BN catalyst surface. This weak interaction favors the rapid desorption of ethylene, thereby preventing further hydrogenation to ethane. The absence of strong anchoring functionality such as conjugated or aromatic systems means that ethylene does not remain on the surface long enough to undergo further reduction. In contrast, as the reviewer noted, our previous study on styrene hydrogenation (DOI: 10.1021/jacs.2c00343) demonstrated that the aromatic phenyl ring significantly enhances adsorption through π - π interactions with the BN surface. This stabilizes the molecule on the catalyst, allowing hydrogenation of the olefinic C=C bond to proceed efficiently, while the aromatic ring remains intact due to its higher resonance stability and relatively weaker interaction with hydrogen species. In the current system, BN defects likely provide FLP-like acid-base sites, which activate H₂ and facilitate selective hydrogenation of the more reactive C≡C bond in acetylene. However, once ethylene is formed, its low binding affinity and lack of specific interactions with the BN surface (e.g., no aromatic anchoring or strong polar groups) result in facile desorption, which kinetically outcompetes further hydrogenation. This explains the observed selectivity toward ethylene formation and highlights the role of defect structure and surface interaction energy in steering product distribution.

(2) Additional DRIFTS experiments under acetylene/H₂ flow

Manuscript (Page 10)

To better understand why hydrogenation terminates at ethylene formation, we performed in situ DRIFTS analysis during acetylene semi-hydrogenation using a gas mixture of 1% C₂H₂/Ar and 5% H₂/Ar (both at 25 ccm; total flow: 50 ccm). Spectra were collected at 25, 100, and 150 °C using BN-700 as the background reference. At 25 °C, new bands appeared at 3254 and 3318 cm⁻¹,^{92,96-98} attributed to physisorbed acetylene molecules, which remained observable upon heating to 100 and 150 °C (Supplementary Fig. 26). Upon increasing the temperature to 100 and 150 °C, additional features emerged. The appearance of N-H stretching bands at 3411 and 3670 cm⁻¹,^{3,92} along with B-H stretching bands at 2498 and 2770 cm⁻¹,^{86,99} confirms that BN-700 effectively activates and dissociates hydrogen (Supplementary Table 3). A notable band at 1202 cm⁻¹, assigned to B-N-H vibrations,¹⁰⁰ further supports this observation. Strong acetylene adsorption via hydrogen bonding to exposed nitrogen sites was also evident through persistent signals at 3254 and 3318 cm⁻¹.^{92,96-98} Importantly, the formation of ethene was detected at elevated temperatures, with its adsorption on the BN surface occurring via weak π -complexation, consistent with a C=C stretching band near 1637, 1756 and 1798 cm⁻¹.^{92,98,101} Notably, no absorption bands were observed around 2900-3000 cm⁻¹,^{102,103} which would indicate the presence of strongly bound =C-H stretches of ethylene or -CH₃ stretches in ethane. This suggests that ethylene, once formed, desorbs readily rather than undergoing further

hydrogenation to ethane, thus explaining the selective termination of hydrogenation at the ethylene stage under the tested reaction conditions.

Supporting Information (Page S6, Supplementary Fig. 26, and Supplementary Table 3)

In-situ acetylene semi-hydrogenation diffuse reflectance infrared Fourier transform spectroscopy (DRIFTS) was performed on a Thermo Fisher Nicolet iS50 FTIR spectrometer equipped with an MCT detector cooled by liquid nitrogen. To begin, the BN-700 sample was pretreated in a Harrick Scientific diffuse reflectance cell equipped with a temperature controller at 150 °C in 50 ccm He for 1 hour. After pretreatment, the background spectra were taken at 25 °C. Next, 1% C₂H₂/Ar at 25 ccm and 5% H₂/Ar at 25 ccm (total flow: 50 ccm) was flowed through the catalyst for 10 min at 25 °C, followed by desorption with 50 ccm He for 10 min. Each spectrum was recorded in 1 min intervals during the adsorption and desorption process with 50 scans at a resolution of 0.4 cm⁻¹. The same procedure as noted above was done for 100 °C and 150 °C where each spectrum was taken without cooling back down to 25 °C.

Supplementary Fig. 26. In-situ acetylene semi-hydrogenation DRIFTS on BN-700 catalyst at 25, 100, and 150 °C.

Supplementary Table 3. Assignment of vibrational bands in the DRIFTS spectra of C₂H₂ adsorption and reaction on BN-700 surface.

Surface Adsorbate	Assignment	Band (cm ⁻¹)	Reference
Weakly absorbed C ₂ H ₂ via H-bond	$\nu_{as}(\text{N-H})$	3670	S41, S42
Acetylene	$\nu(\equiv\text{C-H})$	3300-3200	S41, S43
Acetylide ($-\text{C}\equiv\text{C-H}$)	$\nu_{as}(\text{C-H})$	3250	S43, S44
Surface Boron Hydride	$\nu(\text{B-H}_{\text{ads}})$	2500	S42, S44
Protonated Acetylene H ⁺ (C ₂ H ₂)	$\nu_{as}(\text{C}\equiv\text{C})$	2200	S45
π -bonded CH ₂ =CH ₂	$\nu(\text{C}=\text{C})$	1700-1660	S41, S46 , S47

14) Supplementary: NaNH₂ is extremely corrosive, very few material can withstand molten NaNH₂. Was any nickel incorporated in the structure (ICP-MS analysis?). This could lead to the incorporation of impurities in the BN structure.

Response: We thank the reviewer for the valuable comment. We acknowledge the corrosive nature of molten NaNH₂ and its potential to introduce impurities during synthesis. To assess whether any residual Ni was incorporated into the BN structure, Inductively coupled-plasma optical emission spectroscopy (ICP-OES) analysis was performed on the BN700 catalyst. The results confirmed that no detectable Ni was present, with the detection limit being <0.0009 ppm. This indicates that the h-BN scaffold remained free of Ni contamination under the applied synthesis conditions.

Manuscript (Page 9)

Inductively coupled-plasma optical emission spectroscopy (ICP-OES) analysis of the BN-700 catalyst confirmed the absence of detectable Ni, with a detection limit of <0.0009 ppm, indicating that the BN-700 scaffold remained free of Ni contamination under the applied synthesis conditions.

Supporting Information (Page S5)

Inductively coupled-plasma optical emission spectroscopy (ICP-OES) for the elemental analysis was performed on an Agilent 5110 ICP-OES spectrometer.

15) Supplementary: It should be noted that NaNH_2 reacts vigorously with water. Care must be taken during the washing step.

Response: We thank the reviewer for the valuable comment regarding the safety concerns associated with the use of NaNH_2 . We fully agree that NaNH_2 reacts violently with water, releasing ammonia gas and generating heat, which can pose significant risks if not handled properly. In response to this concern, we have included a detailed description of the synthesis and post-treatment procedures in Supporting Information, along with clear cautionary notes.

Specifically, we emphasize that 1) all handling of NaNH_2 was performed under an inert atmosphere (argon), using standard glovebox to prevent exposure to moisture; 2) The washing step was conducted carefully and gradually, using anhydrous ethanol as the initial quenching medium to safely neutralize residual NaNH_2 before exposure to water. Ethanol reacts more gently with NaNH_2 , minimizing the risk of a violent reaction. 3) After quenching in ethanol, the product was further washed with deionized water to remove any remaining salts, followed by drying under vacuum. These procedures are clearly outlined to ensure reproducibility and safety for future users. We appreciate the reviewer's attention to this important aspect and have taken care to highlight the necessary precautions in the Supporting Information.

Supporting Information (Page S1)

Caution: Sodium amide (NaNH_2) is highly reactive with water, releasing ammonia gas and heat upon contact. To ensure safe handling, all synthesis procedures involving NaNH_2 were conducted under an inert argon atmosphere using a glovebox to avoid moisture exposure. During post-synthesis washing of boron nitride containing trace amounts of NaNH_2 , it is essential to conduct all steps in a well-functioning fume hood to safely vent toxic ammonia gas released during the reaction. The crucible with sample inside should be dropped into plenty of cold water to minimize the reaction rate and manage heat evolution. Appropriate personal protective equipment (PPE) must be worn, including chemical-resistant gloves (e.g., nitrile), safety goggles or a face shield, a lab coat with long sleeves, and respiratory protection if a fume hood is not available (though this is not recommended).

16) Supplementary: What was the pressure of the semihydrogenation reactor? Was the flow 50 ccm, pressure corrected, or 50 sccm, uncorrected

Response: Thank you for your question. The semihydrogenation reactions were carried out at a pressure of 15 psi. To ensure accurate measurement of gas flow under these reaction conditions, we calibrated the flow using an ADM volumetric flow meter, which provides the actual flow rate in ccm rather than relying on standard flow rates (sccm). This approach accounts for deviations in pressure and temperature from standard conditions. Accordingly, the following clarification has been added to the Supporting Information under the "Semihydrogenation Reaction of Acetylene" section. We hope this addition will help eliminate any confusion for readers.

Supporting Information (Page S5)

All semihydrogenation reactions were carried out at a total system pressure of 15 psi. Gas flow rates were calibrated and reported as actual volumetric flow rates (ccm) using an ADM volumetric flow meter, which accounts for deviations from standard temperature and pressure conditions

Reviewer #2 (Remarks to the Author):

The authors reported an impressive work on the fabrication of BN-700 catalyst by the from flux reconstruction method, which exhibited good reactivity in the conversion of semihydrogenation of acetylene in ethylene. The catalyst after the treatment with NaNH₂ was identified with abundant open B, N sites and defects through different characterization technologies, the structure was also simulated to provide more detail information regarding the defects types and stacking models. The catalytic reactivity in activating H₂ was confirmed by the H-D exchange assay, suggesting the high potential in hydrogenation reactions. Appealing, the catalyst was able to hydrogenate a low concentration of acetylene in ethylene, which was an important industrial application, and the reactivity surpass the metal oxide catalyst CeO₂. The catalyst performances was also detailed investigated. The paper was well structure and the data was also comprehensive and convincing. Overall, the paper can be accepted for publication after the minor points are addressed.

1, As for the metal-free catalytic system for acetylene semihydrogenation, the authors could systematically study the literatures and provided the most relevant literatures in this area, to better show the communities the important advance on this specific topic (SI is also fine).

Response: We thank the reviewer for the insightful comments regarding the summary of metal-free catalytic systems for acetylene semihydrogenation. To the best of our knowledge, no widely reported studies exist on acetylene semihydrogenation using metal-free catalysts via thermocatalysis; most literature in this area focuses on noble and transition metal catalysts (DOI: 10.1021/acs.chemrev.9b00230). We agree that a clearer contextual summary would benefit the manuscript. In response, we have included a new table (**Supplementary Table 2**) in the Supporting Information that highlights key advances in FLP-catalyzed reactions involving small gas molecules. The summary spans systems based on boron nitride, heteroatom-doped carbon scaffolds, metal oxides, semi-heterogeneous catalysts, and selected homogeneous FLP systems. The main text has been revised accordingly to reflect this addition.

Manuscript (Page 2)

Notably, the high structural tunability of h-BN enables the controlled formation of unsaturated boron and nitrogen sites in close proximity, which can act as strong acid–base centers to activate small gas molecules via frustrated Lewis pair (FLP)-like behavior (Tables S2). Such mechanisms have been demonstrated in previous studies using h-BN derived from ionothermal synthesis, as well as B- and N-doped carbon materials.⁴²⁻⁴⁶ Furthermore, the activation of small gas molecules via coexisting acid and base sites has also been reported in FLP catalysts based on heteroatom-doped carbon scaffolds,^{47,48} CeO₂,⁴⁹ and InOH,⁵⁰ as well as in semi-heterogeneous^{51,52} and homogeneous systems.^{53,54}

Supporting Information (Page S20, Supplementary Table 2)

Supplementary Table 2 Selected examples of the previously reported heterogenous and homogeneous catalytic FLP systems.

FLP catalyst	Structure	Acid sites	Base sites	Reaction	Ref.
------------------	-------------------	-------------------	-----------------	-------------

Heterogeneous		B sites	N sites	Hydrogenation of alkenes and alkynes	S28
Heterogeneous		B sites	N sites	Hydrogenation of alkenes and alkynes	S29
Heterogeneous		B sites	N sites	Olefin hydrogenation via mechanochemistry	S30
Heterogeneous		B sites	N sites	Selective Hydrogenation	S31
Heterogeneous		B sites	N sites	Electrocatalytic nitrogen reduction	S32
Heterogeneous		C sites	N sites	Dehydrogenation	S33
Heterogeneous		C sites	Heteroatoms	Alkene hydrogenation	S34
Heterogeneous		surface defects (adjacent surface Ce ³⁺)	surface lattice oxygen	Hydrogenation of styrene	S35

Heterogeneous		In(III) sites	InOH sites	Hydrogenation of CO ₂	S36
Semi-heterogeneous		B(C ₆ F ₅) ₃	Oxygen atoms on the surface of the α-cyclodextrin	Hydrogenation, Deoxygenation	S37
Semi-heterogeneous		B(C ₆ F ₅) ₃	Pyridyl sites	Hydrogenation of ketones	S38
Homogeneous		B(C ₆ F ₅) ₃	electron-rich B atoms	Hydrogenation	S39
Homogeneous		2,2,6,6-tetramethylpiperidine (TMP)	B in H-TMP	Hydrogenation of CO ₂ to CH ₃ OH	S40

2, As for the metal-free catalyst, people may care about the existence of trace amount of metals, which sometimes may serve as the catalytic sites for hydrogenation. To verify that, the authors are suggested to carry out the ICP analysis of the relevant catalysts to exclude this.

Response: We thank the reviewer for the valuable comment. To assess whether any residual Ni was incorporated into the BN structure, Inductively coupled-plasma optical emission spectroscopy (ICP-OES) analysis was performed on the BN-700 catalyst. The results confirmed that no detectable Ni was present,

with the detection limit being <0.0009 ppm. This indicates that the h-BN scaffold remained free of Ni contamination under the applied synthesis conditions.

Manuscript (Page 9)

Inductively coupled-plasma optical emission spectroscopy (ICP-OES) analysis of the BN-700 catalyst confirmed the absence of detectable Ni, with a detection limit of <0.0009 ppm, indicating that the BN-700 scaffold remained free of Ni contamination under the applied synthesis conditions.

Supporting Information (Page S5)

Inductively coupled-plasma optical emission spectroscopy (ICP-OES) for the elemental analysis was performed on an Agilent 5110 ICP-OES spectrometer.

3, The side product ethane was also noticed in the reaction, is it possible to get a higher selectivity by optimizing the reaction conditions? Besides, what's the reason or principle for the composition of the mixed gas? Especially the hydrogenation ratio in the mixed gas.

Response: We thank the reviewer for the comments regarding the reaction parameter screening and selectivity. The optimized reaction conditions (75 mg catalyst; gas mixture: 0.5% C₂H₂ / 16% H₂ / 83.5% Ar; flow rate: 50 ccm; gas hourly space velocity (GHSV): 4000 h⁻¹; temperature: 150 °C), leading to around 30% acetylene conversion, were determined after comprehensive screening of reaction parameters, including flow rate, catalyst loading, and reaction temperature. The primary objective was to achieve high acetylene conversion while maintaining ethene selectivity above 95%. As outlined in the Introduction part, the goal of acetylene hydrogenation is to remove trace acetylene impurities from ethene streams to obtain high-purity ethene. In industrial applications, naphtha cracking typically produces ethylene streams containing 0.25–2% acetylene impurities.^{1,2} Reducing acetylene concentrations from ~2% to below 5 ppm is essential yet challenging for ensuring the purity required in downstream processes such as the production of ethylene oxide and polymers. Therefore, as long as the catalyst exhibits sufficient activity for acetylene hydrogenation and maintains high ethene selectivity, moderate conversion levels are acceptable, as purification can be achieved through repeated hydrogenation steps. As demonstrated in our catalysis results, the as-synthesized BN700 catalyst retained its activity and selectivity under conditions mimicking an ethene-rich stream (0.25% C₂H₂/10% C₂H₄/16% H₂/73.5% Ar, 150 °C, GHSV 4000 h⁻¹).

As suggested by the reviewer, the following efforts have been made to clarify the reaction condition screening and further improve the catalytic activity:

(1) Detailed results of the reaction condition screening have been included in the **main manuscript (Page 9) and the Supporting Information (Page S5 and Supplementary Fig. 25)**.

Manuscript (Page 9)

To enhance catalyst performance, variations in catalyst loading and flow rate were systematically investigated to identify optimal conditions for improved activity and ethylene selectivity. Reducing the flow rate to 25 ccm led to an increase in acetylene conversion from 30% to 40%; however, ethylene selectivity

dropped from 98% to 81% (Supplementary Fig. 25A). In a separate experiment, increasing the catalyst loading from 75 mg to 300 mg improved conversion from 30% to 46%, but similarly reduced ethylene selectivity to 76% (Supplementary Fig. 25B). For practical acetylene semihydrogenation, the objective is to remove trace acetylene from ethylene streams while preserving high ethylene purity. It is crucial to maintain ethene selectivity above 95%. Therefore, the optimized reaction conditions selected for further studies were: 75 mg catalyst; gas composition: 0.5% C₂H₂/16% H₂/83.5% Ar; flow rate: 50 ccm; GHSV: 4000 h⁻¹; and reaction temperature: 150 °C.

Supporting Information (Page S5 and Supplementary Fig. 25)

Acetylene semihydrogenation experiments were done using an Altamira instruments 200 catalyst characterization flow reactor (AMI-200). Each acetylene semihydrogenation reaction was carried out with 75 mg of catalyst diluted with 75 mg of quartz and a reactant concentration of 0.5% C₂H₂/16% H₂/83.5% Ar with a flow rate of 50 ccm to give a gas hourly space velocity (GHSV) of 4000 h⁻¹. For reaction parameter screening, either the flow rate (25 ccm) or the catalyst loading (300 mg) was varied, while all other conditions were kept constant.

Supplementary Fig. 25. Screening of reaction parameters for acetylene hydrogenation catalyzed by BN-700. (A) 75 mg of BN-700 catalyst using a 25 ccm flow of 0.5% C₂H₂ / 16% H₂ / 83.5% Ar at 150 °C. (B) 300 mg of BN-700 catalyst using a 50 ccm flow of 0.5% C₂H₂ / 16% H₂ / 83.5% Ar at 150 °C.

(2) Preliminary studies were conducted to further enhance the catalytic performance of BN via Ti doping.

Theoretical simulations indicated that Ti doping in BN could significantly improve H₂ adsorption energy from approximately -0.1 eV to -0.7 eV.³ To investigate this, a Ti-doped BN (Ti-BN) catalyst was synthesized via a modified flux reconstruction method (Figures R1A). Preliminary acetylene hydrogenation tests showed that doping 10% of the Ti sites in BN led to a 12% increase in acetylene conversion compared to the BN700 catalyst under identical conditions. However, this improvement came with a 6% decrease in ethylene selectivity (Figure R1B). These findings suggest that heteroatom doping is a promising strategy to fine-tune the catalytic activity and selectivity of BN. Nonetheless, further development is needed to create metal-free BN catalysts with performance comparable to that of noble- and transition-metal-based systems. Future efforts will focus on exploring alternative dopants (e.g., P and F), optimizing dopant concentrations, tuning porosity, and refining catalyst morphology.

Figure R1 (A) XRD pattern of Ti-BN synthesized through a flux method. (B) C₂H₂ hydrogenation over 10% Ti-doped BN catalyst using 75 mg catalyst with 50 ccm of 0.5% C₂H₂/16% H₂/83.5% Ar mixture over a temperature range of 100 to 150 °C.

(3) Regarding the gas composition, we employed a feed of 0.5% C₂H₂/16% H₂/83.5% Ar, corresponding to a hydrogen-to-acetylene ratio of 32:1, which is far exceeding the stoichiometric requirement (2:1). This high H₂ concentration was chosen to mimic excess hydrogen conditions typical of industrial front-end acetylene hydrogenation, where minimizing residual acetylene is essential for downstream polymerization. The elevated H₂ level also ensures stable catalytic performance and prevents hydrogen starvation under transient conditions. For comparison, we also tested a lower H₂ concentration (2.5%, or a 5:1 H₂/C₂H₂ ratio), under which acetylene conversion dropped to ~5%, confirming that the reaction is strongly hydrogen-limited at reduced H₂ partial pressures and that sufficient hydrogen is critical to initiate and sustain hydrogenation.

4, The releasing of the ethylene from catalytic sites is also important to the high selectivity. Although the authors provided calculation results about the low tendency of ethylene adsorption on catalyst surface. Is it possible to investigate the adsorption behaviors of acetylene and ethylene on catalyst surface and compare that?

Response: We thank the reviewer for raising this insightful point. This question has been addressed through two complementary approaches: theoretical simulations and additional DRIFTS experiments under acetylene/H₂ flow.

(1) Clarification of simulation discussion

The key distinction in the present case of acetylene semihydrogenation lies in the weak adsorption strength of ethylene on the BN catalyst surface. This weak interaction favors the rapid desorption of ethylene, thereby preventing further hydrogenation to ethane. The absence of strong anchoring functionality such as conjugated or aromatic systems means that ethylene does not remain on the surface long enough to undergo further reduction. In contrast, as the reviewer noted, our previous study on styrene hydrogenation (DOI: 10.1021/jacs.2c00343) demonstrated that the aromatic phenyl ring significantly enhances adsorption through π - π interactions with the BN surface. This stabilizes the molecule on the catalyst, allowing hydrogenation of the olefinic C=C bond to proceed efficiently, while the aromatic ring remains intact due to its higher resonance stability and relatively weaker interaction with hydrogen species. In the current system, BN defects likely provide FLP-like acid-base sites, which activate H₂ and facilitate selective hydrogenation of the more reactive C \equiv C bond in acetylene. However, once ethylene is formed, its low binding affinity and lack of specific interactions with the BN surface (e.g., no aromatic anchoring or strong polar groups) result in facile desorption, which kinetically outcompetes further hydrogenation. This explains the observed selectivity toward ethylene formation and highlights the role of defect structure and surface interaction energy in steering product distribution.

(2) Additional DRIFTS experiments under acetylene/H₂ flow

Manuscript (Page 10)

To better understand why hydrogenation terminates at ethylene formation, we performed in situ DRIFTS analysis during acetylene semi-hydrogenation using a gas mixture of 1% C₂H₂/Ar and 5% H₂/Ar (both at 25 ccm; total flow: 50 ccm). Spectra were collected at 25, 100, and 150 °C using BN-700 as the background reference. At 25 °C, new bands appeared at 3254 and 3318 cm⁻¹,^{92,96-98} attributed to physisorbed acetylene molecules, which remained observable upon heating to 100 and 150 °C (Supplementary Fig. 26). Upon increasing the temperature to 100 and 150 °C, additional features emerged. The appearance of N-H stretching bands at 3411 and 3670 cm⁻¹,^{3,92} along with B-H stretching bands at 2498 and 2770 cm⁻¹,^{86,99} confirms that BN-700 effectively activates and dissociates hydrogen (Supplementary Table 3). A notable band at 1202 cm⁻¹, assigned to B-N-H vibrations,¹⁰⁰ further supports this observation. Strong acetylene adsorption via hydrogen bonding to exposed nitrogen sites was also evident through persistent signals at

3254 and 3318 cm^{-1} .^{92,96-98} Importantly, the formation of ethene was detected at elevated temperatures, with its adsorption on the BN surface occurring via weak π -complexation, consistent with a C=C stretching band near 1637, 1756 and 1798 cm^{-1} .^{92,98,101} Notably, no absorption bands were observed around 2900-3000 cm^{-1} ,^{102,103} which would indicate the presence of strongly bound =C-H stretches of ethylene or -CH₃ stretches in ethane. This suggests that ethylene, once formed, desorbs readily rather than undergoing further hydrogenation to ethane, thus explaining the selective termination of hydrogenation at the ethylene stage under the tested reaction conditions.

Supporting Information (Page S6, Supplementary Fig. 26, and Supplementary Table 3)

In-situ acetylene semi-hydrogenation diffuse reflectance infrared Fourier transform spectroscopy (DRIFTS) was performed on a Thermo Fisher Nicolet iS50 FTIR spectrometer equipped with an MCT detector cooled by liquid nitrogen. To begin, the BN-700 sample was pretreated in a Harrick Scientific diffuse reflectance cell equipped with a temperature controller at 150 °C in 50 ccm He for 1 hour. After pretreatment, the background spectra were taken at 25 °C. Next, 1% C₂H₂/Ar at 25 ccm and 5% H₂/Ar at 25 ccm (total flow: 50 ccm) was flowed through the catalyst for 10 min at 25 °C, followed by desorption with 50 ccm He for 10 min. Each spectrum was recorded in 1 min intervals during the adsorption and desorption process with 50 scans at a resolution of 0.4 cm^{-1} . The same procedure as noted above was done for 100 °C and 150 °C where each spectrum was taken without cooling back down to 25 °C.

Supplementary Fig. 26. In-situ acetylene semi-hydrogenation DRIFTS on BN-700 catalyst at 25, 100, and 150 °C.

Supplementary Table 3. Assignment of vibrational bands in the DRIFTS spectra of C₂H₂ adsorption and reaction on BN-700 surface.

Surface Adsorbate	Assignment	Band (cm ⁻¹)	Reference
Weakly absorbed C ₂ H ₂ via H-bond	$\nu_{as}(\text{N-H})$	3670	S41, S42
Acetylene	$\nu(\equiv\text{C-H})$	3300-3200	S41, S43
Acetylide ($-\text{C}\equiv\text{C-H}$)	$\nu_{as}(\text{C-H})$	3250	S43, S44
Surface Boron Hydride	$\nu(\text{B-H}_{\text{ads}})$	2500	S42, S44
Protonated Acetylene H ⁺ (C ₂ H ₂)	$\nu_{as}(\text{C}\equiv\text{C})$	2200	S45
π -bonded CH ₂ =CH ₂	$\nu(\text{C}=\text{C})$	1700-1660	S41, S46 , S47

5, A schematic catalytic cycle of the reaction pathway could be provided to better show the reaction mechanism.

Response: We thank the reviewer for the helpful suggestion. To better illustrate the proposed reaction mechanism, we have added a schematic catalytic cycle.

Manuscript (Page 11)

Based on the simulation and DRIFTS observations, a schematic catalytic cycle is proposed (Supplementary Fig. 27).

Supporting Information (Page S18 and Supplementary Fig. 27)

Supplementary Fig. 27. Proposed acetylene semi-hydrogenation mechanism based on BN model with BN_2 defect.

Reviewer #3 (Remarks to the Author):

Ethylene purification remains a long-standing challenge in various industrial processes where ethylene serves as a fundamental building block. Conventional catalysts for selective acetylene hydrogenation primarily rely on noble and transition metal active sites, which are constrained by high costs and poor thermal stability. In this work, Dai et al. introduce a novel metal-free catalyst for the highly selective semihydrogenation of acetylene in ethylene streams. The defect-rich boron nitride (BN) catalyst was synthesized via flux reconstruction, and its structural properties, including defect types and stacking modes, were comprehensively characterized. BN-700 exhibited remarkable H_2 dissociation activity in both liquid and solid phases, leading to highly selective acetylene semihydrogenation. The low energy barrier for H_2 activation, coupled with accelerated ethylene desorption to prevent over-hydrogenation, was validated through isotope studies and theoretical simulations. Given the well-structured presentation and significant scientific contribution, I recommend acceptance of this manuscript after minor revisions.

1. The flux reconstruction process for converting AMBN into BN nanoparticles is impressive, particularly in achieving the removal of O and C impurities. However, it remains unclear where these impurities go. A TG-MS (thermogravimetric-mass spectrometry) experiment is recommended to analyze the gaseous byproducts and provide insights into the fate of O and C impurities.

Response: Thanks for the great suggestions. We have conducted TG-MS tests and added a new figure in the SI. As shown in Supplementary Fig. ##, the weight loss of the AMBN/ NaNH_2 mixture around 200°C is primarily due to the release of NH_3 , H_2 , and H_2O . As the temperature increases to 700°C , the MS signals of CO_2 become more pronounced, indicating its release at higher temperatures. Thus, the main gaseous byproducts during flux reconstruction are NH_3 , H_2O , H_2 , and CO_2 . The O and C impurities in AMBN are converted into H_2O and CO_2 . We have added the experiment details in supporting information and the related discussion in main text.

Manuscript (Page 6)

Thermogravimetric-mass spectrometry (TG-MS) tests were conducted to analyze the gas products released during flux reconstruction. As shown in Supplementary Fig. 7, the weight loss of the AMBN/ NaNH_2 mixture around 200 °C is primarily due to the release of NH_3 , H_2 , and H_2O . As the temperature increases to 700 °C, the MS signals of CO_2 become more pronounced, indicating its release at higher temperatures. Thus, the main gaseous byproducts during flux reconstruction are NH_3 , H_2O , H_2 , and CO_2 . The O and C impurities in AMBN are converted into H_2O and CO_2 .

Supporting Information (Page S2 and Supplementary Fig. 7)

Supplementary Fig. 7. TG-MS results of AMBN/ NaNH_2 mixture from room temperature to 700 °C.

Thermogravimetric-mass spectrometry (TG-MS) tests were conducted using a Discovery TGA-MS under a nitrogen (N_2) atmosphere. The heating protocol involved heating the AMBN/ NaNH_2 mixture from room temperature to 700 °C at a rate of 5 °C/min, followed by a 30-minute hold at 700 °C.

2. While the manuscript maintains consistent terminology, some technical terms require clearer definitions for broader readability. For instance, the terms " BN_2 vacancy" and " BN_3 vacancy" should be explicitly defined in the introduction or methods section to assist non-expert readers.

Response: We have added illustrations of the BN_2 and BN_3 vacancies in Supplementary Fig. 10 to help non-expert readers understand the different types of vacancies in BN.

Supporting Information (Page S10 and Figure 10)

Supplementary Fig. 10. BN models with different defect structures based on theoretical study from previous work.^{S17}

3. The statement "The AMBN precursor displayed the presence of tetracoordinated $\text{BN}_x\text{O}_{4-x}$ ($x = 0-4$) species, together with the regular tricoordinate BN_3 bonds (Figure 2a)" introduces the concept of tetracoordinated $\text{BN}_x\text{O}_{4-x}$ species, which may be unfamiliar to some readers. While the BN_3 bonds are well understood, an illustration or reference supporting the tetracoordinated $\text{BN}_x\text{O}_{4-x}$ structure would enhance clarity.

Response: Thanks for your suggestion. We have provided an illustration to show $\text{BN}_x\text{O}_{4-x}$ species in supporting information.

Supporting Information (Page S8 and Supplementary Fig. 6)

Supplementary Fig. 6. Illustration of tricoordinate BN_3 bonds and tetracoordinated $\text{BN}_x\text{O}_{4-x}$ ($x = 0-3$) species.

4. The DFT calculations effectively support the experimental findings. However, it would be beneficial to compare the computed H_2 activation energy with previously reported catalysts to contextualize the performance of BN-700 relative to existing materials.

Response: Great suggestion! We have compared the computed H_2 activation energy with reported catalysts and summarized in a table supporting information. The H_2 activation energy on BN-700 is lower than pristine CeO_2 and TiO_2 , and comparable to T-H600 and model molecular $\text{BH}_2\text{CH}_2\text{NH}_2$.

Supplementary Table 4. Comparison of H_2 activation energy on different noble-metal free catalysts.

Catalyst	Material	H ₂ activation energy (kJ/mol)	Reference
BN-700	BN	58.2	This work
CeO₂	CeO ₂	70.7	This work
CeO₂	CeO ₂	96	S48
BH₂CH₂NH₂	model molecule	50.2	S49
BH₂NH₂	model molecule	178.6	S49
BH₂PH₂	model molecule	77.4	S49
T-H600	TiO ₂	56.4	S50
TiO₂-fresh catalyst	TiO ₂	72	S50
Cu₁/PHI	Cu on poly(heptazine imide)	70.9	S51

5. Some recent progress in selective semihydrogenation should be cited, including J. Am. Chem. Soc. 2021, 143, 4483, Angew. Chem. Int. Ed. 2024,63, e202410394, which should enhance the manuscript's readability.

Response: Thanks for your suggestion. We have cited and discussed the related references in introduction section to enhance the manuscript's readability.

State-of-the-art catalysts in selective acetylene hydrogenation mainly involve noble- and transition-metal nanoparticles (NPs), alloys, or single atoms (SAs, e.g., Pd, Au, Cu, and Ni),⁸⁻¹⁰ with the active sites being isolated via metal surface modification,¹¹ strong metal-support interaction (SMSI) construction,¹² and confinement effect introduction by porosity control to ensure high selectivity towards ethylene production.^{3,13,14} Despite significant progress, the development of metal-free catalysts remains highly attractive due to the limited natural abundance of platinum group metals (PGMs) and the persistent challenges of sintering and coking in metal-catalyzed hydrogenation processes.

References:

- Zhang, L., Zhou, M., Wang, A. & Zhang, T. Selective Hydrogenation over Supported Metal Catalysts: From Nanoparticles to Single Atoms. *Chem. Rev.* **120**, 683-733 (2020). <https://doi.org/10.1021/acs.chemrev.9b00230>
- Studt, F. *et al.* Identification of Non-Precious Metal Alloy Catalysts for Selective Hydrogenation of Acetylene. *Science* **320**, 1320-1322 (2008). <https://doi.org/doi:10.1126/science.1156660>
- Zhang, L., Lin, J., Liu, Z. & Zhang, J. Non-noble metal-based catalysts for acetylene semihydrogenation: from thermocatalysis to sustainable catalysis. *Sci. China Chem.* **66**, 1963-1974 (2023). <https://doi.org/10.1007/s11426-022-1597-y>
- Huang, F. *et al.* Anchoring Cu₁ species over nanodiamond-graphene for semi-hydrogenation of acetylene. *Nat. Commun.* **10**, 4431 (2019). <https://doi.org/10.1038/s41467-019-12460-7>
- Liu, K., Qin, R. & Zheng, N. Insights into the interfacial effects in heterogeneous metal nanocatalysts toward selective hydrogenation. *Journal of the American Chemical Society* **143**, 4483-4499 (2021).
- Yang, B. *et al.* Incorporation of Pd Single-Atom Sites in Perovskite with an Excellent Selectivity toward Photocatalytic Semihydrogenation of Alkynes. *Angew. Chem. Int. Ed.* **63**, e202410394 (2024).

- 13 Li, Y. *et al.* Mechanistic and Atomic-Level Insights into Semihydrogenation Catalysis to Light Olefins. *ACS Catal.* **12**, 12138-12161 (2022). <https://doi.org/10.1021/acscatal.2c03750>
- 14 Zou, S. *et al.* Grafting nanometer metal/oxide interface towards enhanced low-temperature acetylene semi-hydrogenation. *Nat. Commun.* **12**, 5770 (2021). <https://doi.org/10.1038/s41467-021-25984-8>
- 36 Jhi, S.-H. & Kwon, Y.-K. Hydrogen adsorption on boron nitride nanotubes: A path to room-temperature hydrogen storage. *Phys. Rev. B* **69** (2004). <https://doi.org/10.1103/PhysRevB.69.245407>
- 37 Wu, X., Yang, J., Hou, J. G. & Zhu, Q. Defects-enhanced dissociation of H₂ on boron nitride nanotubes. *J. Chem. Phys.* **124**, 054706 (2006). <https://doi.org/10.1063/1.2162897>
- 38 Shevlin, S. A. & Guo, Z. X. Hydrogen sorption in defective hexagonal BN sheets and BN nanotubes. *Phys. Rev. B* **76** (2007). <https://doi.org/10.1103/PhysRevB.76.024104>
- 39 Chen, H. *et al.* Defect-Regulated Frustrated-Lewis SI. *J. Am. Chem. Soc.* **144**, 10688-10693 (2022). <https://doi.org/10.1021/jacs.2c00343>
- 40 Sassi, M. & Autrey, T. First-Principles Study of Molecular Hydrogen Activation by Defects in Boron Nitride. *J. Phys. Chem. C* **129**, 6657-6665 (2025). <https://doi.org/10.1021/acs.jpcc.5c00806>
- 42 Li, M. *et al.* Construction of Boron- and Nitrogen-Enriched Nanoporous pi-Conjugated Networks Towards Enhanced Hydrogen Activation. *Angew. Chem. Int. Ed.* **62**, e202302684 (2023). <https://doi.org/10.1002/anie.202302684>
- 43 Chen, H. *et al.* Defect-Regulated Frustrated-Lewis-Pair Behavior of Boron Nitride in Ambient Pressure Hydrogen Activation. *Journal of the American Chemical Society* **144**, 10688-10693 (2022). <https://doi.org/10.1021/jacs.2c00343>
- 44 Ding, Y. *et al.* A Heterogeneous Metal-Free Catalyst for Hydrogenation: Lewis Acid–Base Pairs Integrated into a Carbon Lattice. *Angew. Chem. Int. Ed.* **57**, 13800-13804 (2018). <https://doi.org/https://doi.org/10.1002/anie.201803977>
- 45 Lin, W. *et al.* Creating Frustrated Lewis Pairs in Defective Boron Carbon Nitride for Electrocatalytic Nitrogen Reduction to Ammonia. *Angew. Chem. Int. Ed.* **61**, e202207807 (2022). <https://doi.org/https://doi.org/10.1002/anie.202207807>
- 46 Nash, D. J. *et al.* Heterogeneous Metal-Free Hydrogenation over Defect-Laden Hexagonal Boron Nitride. *ACS Omega* **1**, 1343-1354 (2016). <https://doi.org/10.1021/acsomega.6b00315>
- 47 Primo, A., Neatu, F., Florea, M., Parvulescu, V. & Garcia, H. Graphenes in the absence of metals as carbocatalysts for selective acetylene hydrogenation and alkene hydrogenation. *Nat. Commun.* **5**, 5291 (2014). <https://doi.org/10.1038/ncomms6291>
- 48 Hu, H. *et al.* Metal-free carbocatalyst for room temperature acceptorless dehydrogenation of N-heterocycles. *Sci. Adv.* **8**, eabl9478 (2022). <https://doi.org/10.1126/sciadv.abl9478>
- 49 Zhang, S. *et al.* Solid frustrated-Lewis-pair catalysts constructed by regulations on surface defects of porous nanorods of CeO₂. *Nat. Commun.* **8**, 15266 (2017). <https://doi.org/10.1038/ncomms15266>
- 50 Ghuman, K. K. *et al.* Surface Analogues of Molecular Frustrated Lewis Pairs in Heterogeneous CO₂ Hydrogenation Catalysis. *ACS Catal.* **6**, 5764-5770 (2016). <https://doi.org/10.1021/acscatal.6b01015>

- 51 Mahdi, T. & Stephan, D. W. Facile Protocol for Catalytic Frustrated Lewis Pair Hydrogenation and Reductive Deoxygenation of Ketones and Aldehydes. *Angew. Chem. Int. Ed.* **54**, 8511-8514 (2015). <https://doi.org/https://doi.org/10.1002/anie.201503087>
- 52 Tian, C. *et al.* Use of steric encumbrance to develop conjugated nanoporous polymers for metal-free catalytic hydrogenation. *Chem. Commun.* **52**, 11919-11922 (2016). <https://doi.org/10.1039/C6CC06372A>
- 53 Welch, G. C., Juan, R. R. S., Masuda, J. D. & Stephan, D. W. Reversible, Metal-Free Hydrogen Activation. *Science* **314**, 1124-1126 (2006). <https://doi.org/10.1126/science.1134230>
- 54 Ashley, A. E., Thompson, A. L. & O'Hare, D. Non-Metal-Mediated Homogeneous Hydrogenation of CO₂ to CH₃OH. *Angew. Chem. Int. Ed.* **48**, 9839-9843 (2009). <https://doi.org/10.1002/anie.200905466>
- 56 Wu, P. *et al.* A template-free solvent-mediated synthesis of high surface area boron nitride nanosheets for aerobic oxidative desulfurization. *Chem. Commun.* **52**, 144-147 (2016). <https://doi.org/10.1039/C5CC07830J>
- 57 Marchesini, S., McGilvery, C. M., Bailey, J. & Petit, C. Template-Free Synthesis of Highly Porous Boron Nitride: Insights into Pore Network Design and Impact on Gas Sorption. *ACS Nano* **11**, 10003-10011 (2017). <https://doi.org/10.1021/acsnano.7b04219>
- 58 Zhang, N. *et al.* The Research of the Synthesis Mechanism and Synthesis Process of High Crystallinity Globular h-BN. *J. Inorg. Organomet. Polym. Mater.* **25**, 1495-1501 (2015). <https://doi.org/10.1007/s10904-015-0268-4>
- 68 Chen, H. *et al.* Defect-Regulated Frustrated-Lewis-Pair Behavior of Boron Nitride in Ambient Pressure Hydrogen Activation. *J. Am. Chem. Soc.* **144**, 10688-10693 (2022). <https://doi.org/10.1021/jacs.2c00343>
- 70 Rabiei, M. *et al.* X-ray Diffraction Analysis and Williamson-Hall Method in USDM Model for Estimating More Accurate Values of Stress-Strain of Unit Cell and Super Cells (2 × 2 × 2) of Hydroxyapatite, Confirmed by Ultrasonic Pulse-Echo Test. *Materials (Basel)* **14** (2021). <https://doi.org/10.3390/ma14112949>
- 92 Moon, J. *et al.* Discriminating the role of surface hydride and hydroxyl for acetylene semihydrogenation over ceria through in situ neutron and infrared spectroscopy. *ACS Catal.* **10**, 5278-5287 (2020).
- 93 Li, M. *et al.* Construction of Boron-and Nitrogen-Enriched Nanoporous π -Conjugated Networks Towards Enhanced Hydrogen Activation. *Angew. Chem. Int. Ed.* **135**, e202302684 (2023).
- 94 Pei, G. X. *et al.* Performance of Cu-alloyed Pd single-atom catalyst for semihydrogenation of acetylene under simulated front-end conditions. *ACS Catal.* **7**, 1491-1500 (2017).
- 95 Xie, K. *et al.* Catalysts for selective hydrogenation of acetylene: A review. *Materials Today Catalysis*, 100029 (2023).
- 96 Ivanov, A. V., Koklin, A. E., Uvarova, E. B. & Kustov, L. M. A DRIFT spectroscopic study of acetylene adsorbed on metal oxides. *Physical Chemistry Chemical Physics* **5**, 4718-4723 (2003).
- 97 Douberly, G. E. *et al.* Infrared photodissociation spectroscopy of protonated acetylene and its clusters. *J. Phys. Chem. A* **112**, 1897-1906 (2008).

- 98 Cao, T. *et al.* An in situ DRIFTS mechanistic study of CeO₂-catalyzed acetylene semihydrogenation reaction. *Physical Chemistry Chemical Physics* **20**, 9659-9670 (2018).
- 99 Tominaka, S. *et al.* Geometrical frustration of BH bonds in layered hydrogen borides accessible by soft chemistry. *Chem* **6**, 406-418 (2020).
- 100 Yang, J. *et al.* In situ IR comparative study on N₂O formation pathways over different valence states manganese oxides catalysts during NH₃-SCR of NO. *Chem. Eng. J.* **397**, 125446 (2020). <https://doi.org/https://doi.org/10.1016/j.cej.2020.125446>
- 101 Carrasco, J. *et al.* Molecular-level understanding of CeO₂ as a catalyst for partial alkyne hydrogenation. *J. Phys. Chem. C* **118**, 5352-5360 (2014).
- 102 Abreu, N. J., Valdés, H., Zaror, C. A., Azzolina-Jury, F. & Meléndrez, M. F. Ethylene adsorption onto natural and transition metal modified Chilean zeolite: An operando DRIFTS approach. *Microporous Mesoporous Mater.* **274**, 138-148 (2019). <https://doi.org/https://doi.org/10.1016/j.micromeso.2018.07.043>
- 103 Pidko, E. & Kazansky, V. σ -Type ethane adsorption complexes with Cu⁺ ions in Cu(i)-ZSM-5 zeolite. Combined DRIFTS and DFT study. *Physical Chemistry Chemical Physics* **7**, 1939-1944 (2005). <https://doi.org/10.1039/B418498J>
- S18 Jhi, S.-H. & Kwon, Y.-K. Hydrogen adsorption on boron nitride nanotubes: A path to room-temperature hydrogen storage. *Phys. Rev. B* **69** (2004). <https://doi.org/10.1103/PhysRevB.69.245407>
- S19 Mpourmpakis, G. & Froudakis, G. E. Why boron nitride nanotubes are preferable to carbon nanotubes for hydrogen storage?: An ab initio theoretical study. *Catalysis Today* **120**, 341-345 (2007). <https://doi.org/https://doi.org/10.1016/j.cattod.2006.09.023>
- S20 Wu, X., Yang, J., Hou, J. G. & Zhu, Q. Defects-enhanced dissociation of H₂ on boron nitride nanotubes. *J. Chem. Phys.* **124**, 054706 (2006). <https://doi.org/10.1063/1.2162897>
- S21 Shevlin, S. A. & Guo, Z. X. Hydrogen sorption in defective hexagonal BN sheets and BN nanotubes. *Phys. Rev. B* **76** (2007). <https://doi.org/10.1103/PhysRevB.76.024104>
- S22 Chen, H. *et al.* Defect-Regulated Frustrated-Lewis SI. *J. Am. Chem. Soc.* **144**, 10688-10693 (2022). <https://doi.org/10.1021/jacs.2c00343>
- S23 Sassi, M. & Autrey, T. First-Principles Study of Molecular Hydrogen Activation by Defects in Boron Nitride. *J. Phys. Chem. C* **129**, 6657-6665 (2025). <https://doi.org/10.1021/acs.jpcc.5c00806>
- S24 Ghosh, S., Nath, P., Moshat, S. & Sanyal, D. Role of carbon substitutional and vacancy in tailoring the H₂ adsorption energy over a hexagonal boron nitride monolayer: an ab initio study. *Journal of Materials Science* **59**, 10877-10887 (2024). <https://doi.org/10.1007/s10853-024-09807-x>
- S25 Panigrahi, P. *et al.* Capacity enhancement of polyolithiated functionalized boron nitride nanotubes: an efficient hydrogen storage medium. *Phys Chem Chem Phys* **22**, 15675-15682 (2020). <https://doi.org/10.1039/d0cp01237h>
- S26 Bi, L., Yin, J., Huang, X., Wang, Y. & Yang, Z. A DFT study of H₂ adsorption on lithium decorated 3D hybrid Boron-Nitride-Carbon frameworks. *International Journal of Hydrogen Energy* **44**, 15183-15192 (2019). <https://doi.org/10.1016/j.ijhydene.2019.04.114>
- S27 Shevlin, S. A. & Guo, Z. X. Transition-metal-doping-enhanced hydrogen storage in boron nitride systems. *Applied Physics Letters* **89** (2006). <https://doi.org/10.1063/1.2360232>

- S28 Li, M. *et al.* Construction of Boron- and Nitrogen-Enriched Nanoporous pi-Conjugated Networks Towards Enhanced Hydrogen Activation. *Angew. Chem. Int. Ed.* **62**, e202302684 (2023). <https://doi.org/10.1002/anie.202302684>
- S29 Chen, H. *et al.* Defect-Regulated Frustrated-Lewis-Pair Behavior of Boron Nitride in Ambient Pressure Hydrogen Activation. *Journal of the American Chemical Society* **144**, 10688-10693 (2022). <https://doi.org/10.1021/jacs.2c00343>
- S30 Nash, D. J. *et al.* Heterogeneous Metal-Free Hydrogenation over Defect-Laden Hexagonal Boron Nitride. *ACS Omega* **1**, 1343-1354 (2016). <https://doi.org/10.1021/acsomega.6b00315>
- S31 Ding, Y. *et al.* A Heterogeneous Metal-Free Catalyst for Hydrogenation: Lewis Acid-Base Pairs Integrated into a Carbon Lattice. *Angew. Chem. Int. Ed.* **57**, 13800-13804 (2018). <https://doi.org/https://doi.org/10.1002/anie.201803977>
- S32 Lin, W. *et al.* Creating Frustrated Lewis Pairs in Defective Boron Carbon Nitride for Electrocatalytic Nitrogen Reduction to Ammonia. *Angew. Chem. Int. Ed.* **61**, e202207807 (2022). <https://doi.org/https://doi.org/10.1002/anie.202207807>
- S33 Hu, H. *et al.* Metal-free carbocatalyst for room temperature acceptorless dehydrogenation of N-heterocycles. *Sci. Adv.* **8**, eabl9478 (2022). <https://doi.org/10.1126/sciadv.abl9478>
- S34 Primo, A., Neatu, F., Florea, M., Parvulescu, V. & Garcia, H. Graphenes in the absence of metals as carbocatalysts for selective acetylene hydrogenation and alkene hydrogenation. *Nat. Commun.* **5**, 5291 (2014). <https://doi.org/10.1038/ncomms6291>
- S35 Zhang, S. *et al.* Solid frustrated-Lewis-pair catalysts constructed by regulations on surface defects of porous nanorods of CeO₂. *Nat. Commun.* **8**, 15266 (2017). <https://doi.org/10.1038/ncomms15266>
- S36 Ghuman, K. K. *et al.* Surface Analogues of Molecular Frustrated Lewis Pairs in Heterogeneous CO₂ Hydrogenation Catalysis. *ACS Catal.* **6**, 5764-5770 (2016). <https://doi.org/10.1021/acscatal.6b01015>
- S37 Mahdi, T. & Stephan, D. W. Facile Protocol for Catalytic Frustrated Lewis Pair Hydrogenation and Reductive Deoxygenation of Ketones and Aldehydes. *Angew. Chem. Int. Ed.* **54**, 8511-8514 (2015). <https://doi.org/https://doi.org/10.1002/anie.201503087>
- S38 Tian, C. *et al.* Use of steric encumbrance to develop conjugated nanoporous polymers for metal-free catalytic hydrogenation. *Chem. Commun.* **52**, 11919-11922 (2016). <https://doi.org/10.1039/C6CC06372A>
- S39 Welch, G. C., Juan, R. R. S., Masuda, J. D. & Stephan, D. W. Reversible, Metal-Free Hydrogen Activation. *Science* **314**, 1124-1126 (2006). <https://doi.org/10.1126/science.1134230>
- S40 Ashley, A. E., Thompson, A. L. & O'Hare, D. Non-Metal-Mediated Homogeneous Hydrogenation of CO₂ to CH₃OH. *Angew. Chem. Int. Ed.* **48**, 9839-9843 (2009). <https://doi.org/10.1002/anie.200905466>
- S41 Moon, J. *et al.* Discriminating the role of surface hydride and hydroxyl for acetylene semihydrogenation over ceria through in situ neutron and infrared spectroscopy. *ACS Catal.* **10**, 5278-5287 (2020).
- S42 Chen, H. *et al.* Defect-regulated Frustrated-Lewis-Pair behavior of boron nitride in ambient pressure hydrogen activation. *J. Am. Chem. Soc.* **144**, 10688-10693 (2022).

- S43 Ivanov, A. V., Koklin, A. E., Uvarova, E. B. & Kustov, L. M. A DRIFT spectroscopic study of acetylene adsorbed on metal oxides. *Physical Chemistry Chemical Physics* **5**, 4718-4723 (2003).
- S44 Tominaka, S. *et al.* Geometrical frustration of BH bonds in layered hydrogen borides accessible by soft chemistry. *Chem* **6**, 406-418 (2020).
- S45 Douberly, G. E. *et al.* Infrared photodissociation spectroscopy of protonated acetylene and its clusters. *J. Phys. Chem. A* **112**, 1897-1906 (2008).
- S46 Cao, T. *et al.* An in situ DRIFTS mechanistic study of CeO₂-catalyzed acetylene semihydrogenation reaction. *Physical Chemistry Chemical Physics* **20**, 9659-9670 (2018).
- S47 Carrasco, J. *et al.* Molecular-level understanding of CeO₂ as a catalyst for partial alkyne hydrogenation. *J. Phys. Chem. C* **118**, 5352-5360 (2014).
- S48 Vilé, G. *et al.* Promoted ceria catalysts for alkyne semi-hydrogenation. *J. Catal.* **324**, 69-78 (2015).
- S49 Wang, Z., Lu, G., Li, H. & Zhao, L. Encumbering the intramolecular π donation by using a bridge: A strategy for designing metal-free compounds to hydrogen activation. *Chin. Sci. Bull.* **55**, 239-245 (2010).
- S50 Wan, Q., Chen, Y., Zhou, S., Lin, J. & Lin, S. Selective hydrogenation of acetylene to ethylene on anatase TiO₂ through first-principles studies. *J. Mater. Chem. A* **9**, 14064-14073 (2021).
- S51 Chen, X., Li, Y., Yuan, Y. & Lin, W. What size of Cu_n clusters loaded on poly(heptazine imide) have better catalytic performance for acetylene semi-hydrogenation? *Mol. Catal.* **569**, 114605 (2024). <https://doi.org/10.1016/j.mcat.2024.114605>

Responses to reviewer comments

Reviewer #1:

The authors have not adequately addressed the issues brought up in the previous review. In particular, this manuscript inappropriately cites previous olefin hydrogenation work over defects in boron nitride (reference 46). This is especially important as key points in this manuscript are not supported by theoretical and experimental results in the prior work.

Response: We sincerely thank the reviewer for highlighting the importance of properly citing and discussing reference 46 (*ACS Omega* 2016, 1, 1343–1354). This work pioneered the heterogeneous metal-free hydrogenation over defect-laden h-BN. As this point was raised in multiple subsequent comments related to hydrogen molecule/atom adsorption energy and hydrogenation reaction pathway, we have carefully addressed the relevant concerns pertaining to this reference in our detailed responses to each comment.

1) A conversion of at most 40% will not reduce a stream containing 0.25-2% down to 0.00005%. That would require a conversion of 99.98%. It seems the path forward will be challenging as efforts to increase conversion led to subsequent reduction in selectivity.

Response: We thank the reviewer for this comment and fully agree that a single-pass conversion of 40% is insufficient to reduce acetylene concentrations from 0.25-2% to sub-ppm levels. In our manuscript, we stated that “In industrial applications, reducing trace acetylene in ethene streams (typically from 1% to below 5 ppm) is essential and highly challenging for maintaining high-purity ethylene in cracking processes.” We would like to clarify that we did not claim BN-700 can directly achieve this industrial target, which remains a grand challenge even for noble metal catalysts. In fact, even industrially implemented noble-metal catalysts (e.g., Pd-based systems) do not achieve such high purity in a single pass. Instead, industrial hydrogenation processes employ multi-bed or staged reactor configurations with precise temperature and hydrogen flow control to achieve both high conversion and ultra-high ethylene selectivity. These reactor engineering strategies are not readily replicated in laboratory-scale flow systems.

Our aim in this work is to explore the fundamental insights of a metal-free, defect-rich BN catalyst to selectively hydrogenate acetylene with minimal over-hydrogenation, while providing a platform for future optimization. Compared to other non-noble-metal catalysts such as Ni-, Cu-, and CeO₂-based systems, our catalyst offers key advantages including the exceptional acetylene-to-ethylene selectivity (>98%), which is among the highest reported for metal-free materials, and complete absence of metallic elements, eliminating concerns over metal site poisoning or sintering.

Moreover, the weak interaction between ethylene and the BN surface, dictated by the electronic structure and defect geometry, favors rapid desorption of the desired product and helps suppress further hydrogenation to ethane. While conversion remains moderate under our lab-scale conditions, this work establishes a promising foundation for future development of metal-free semihydrogenation catalysts.

Titanium doping is presented in the reviewer response but is not in the manuscript materials. Is this supplied as extra information for the reviewers? It seems to introduce more questions than answers. Specifically how the authors could be sure titanium is in the structure of the BN and not

converted to the oxide during removal of the NaNH₂.

Response: We appreciate the reviewer's careful observation. Yes, this information was exploratory and included only to show proof-of-concept for activity tuning via heteroatom doping. This exploratory result is not central to the current study and has therefore not been included in the main manuscript.

The development of Ti-doped BN catalysts is part of our ongoing effort to optimize metal-free systems for semihydrogenation applications and will be the subject of a separate, dedicated study. As the reviewer rightly noted, confirming the incorporation of Ti into the BN lattice (high-resolution forming titanium oxide species) would require advanced characterization techniques such as Titanium K-edge X-ray Absorption Near-Edge Structure (XANES) spectroscopy, X-ray Pair Distribution Function (PDF) analysis, and high resolution microscopy, which were beyond the scope of this work.

2,3)

As the authors point out the conversion from eV to kJ requires proper conversion. Attributing the discrepancy between their values of -505 kJ/mol and -907 kJ/mol and literature values neglects the fact that these values convert to -5.23 eV and -9.40 eV; far stronger binding than most of the values they presented in table S2.

Response: We thank the reviewer for the careful analysis and the opportunity to clarify this point. First, we would like to emphasize that the high adsorption energy values referenced (e.g., -505 kJ/mol and -907 kJ/mol, equivalent to -5.23 eV and -9.40 eV) were taken directly from the literature (particularly Reference S20), and not derived from our own simulations.

These reported values correspond to **atomic hydrogen** adsorption on BN nanotubes with **vacancy defects**, which inherently exhibit stronger binding interactions than molecular hydrogen on planar, defect-free BN surfaces. The large spread in values reported in Table S1 (from -0.037 to -9.40 eV) reflects differences in **the type of adsorbate** (H atom vs. H₂ molecule), **the dimensionality** of the system (nanotube vs. sheet), and **the defect sites** (single vacancy, divacancy, multi-vacancy).

To improve clarity and avoid confusion, we have revised Table S1 to explicitly distinguish between molecular and atomic hydrogen adsorption, and have clarified the dimensional context (e.g., nanotube vs. sheet). The discussion in the Introduction has also been modified accordingly to prevent overgeneralization and misinterpretation of the binding strength trends.

“For example, in perfect sp²-hybridized B–N hexagonal units, the binding energies of H₂ molecules were calculated to be -0.085 eV and -0.1 eV at the B and N sites, respectively. Theoretical simulations indicate that BN scaffolds with vacancies deviating from the ideal B₃N₃ structure can activate and dissociate H₂ molecules through the formation of N–H and B–H bonds, with the adsorption energy of hydrogen atoms as low as -9.40 eV, depending on the vacancy size and atomic configuration (Supplementary Table 1).”

Supplementary Table 1 Summary of H₂ adsorption energies on various boron nitride materials based on theoretical simulations.

Sample	Defect types	Absorbate	Energy type	Value from simulation	Reference
(10,0) boron nitride Nanotubes	B sites and N sites (no defects)	Hydrogen molecule	Adsorption energy	-0.085 and -0.1 eV	S18
(9,9) armchair-type boron nitride nanotubes	B sites and N sites (no defects)	Hydrogen molecule	Adsorption energy	-0.027 and -0.037 eV	S19
boron nitride nanotubes	V_N, V_B	Hydrogen atoms	Adsorption energy	-5.24 and -9.40 eV	S20
hexagonal boron nitride	V_N, V_B	Hydrogen atoms	Adsorption energy	-1.55 and -5.18 eV	S21
hexagonal boron nitride ^[a]	BN, BN ₂ , BN ₃ , B ₃ N ₃ , B ₇ N ₆ , B ₇ N ₁₂	Hydrogen atoms	Dissociative H ₂ adsorption energy	-5.1 to 0.5 eV	S22
hexagonal boron nitride ^[b]	2V(1B1N), 3V(2B1N), 3V(1B2N), 4V(3B1N), 6V(3B3N)	Hydrogen atoms	Gibbs free energy of hydrogenation	-4.0 to -0.5 eV	S23
Carbon-doped hexagonal boron nitride ^[c]	C _B (N,N,N), C _B (C,N,N), C _B (C,C,N), C _B (C,C,C), C _N (B,B,B), C _N (C,B,B), C _N (C,C,B), C _N (C,C,C)	Hydrogen molecule	Adsorption energy	-0.084 to -0.063 eV	S24
Li ⁺ doped boron nitride nanotubes	NA	Hydrogen molecule	Adsorption energy	-0.33 eV	S25
lithium decorated 3D hybrid Boron-Nitride-Carbon frameworks	NA	Hydrogen molecule	Adsorption energy	-0.24 eV	S26
Ti-doped boron nitride	NA	Hydrogen molecule	Adsorption energy	-0.70 eV	S27
	[a] Structure of boron nitride with defects shown in Supplementary Fig. 10.				
	[b] Structure of boron nitride with defects.				

Minor discrepancies are to be expected with different researchers using slightly different parameters. However a factor of more than 2 suggests an error unless the authors have some overwhelming evidence that previous work was performed in an incomplete manner.

Response: We appreciate the reviewer's point regarding the magnitude of discrepancies in reported adsorption energies. Indeed, as illustrated in the first two entries of Table S1, such as -0.085 and -0.1 eV in Reference S18 versus -0.027 and -0.037 eV in Reference S19, minor variations are observed due to differences in computational parameters, defect models, or exchange–correlation functionals.

The significantly more negative values (e.g., < -4 eV) reported in References S20–S22 correspond to **atomic hydrogen adsorption**, often on **defects** in hexagonal BN and BN nanotubes, **rather than molecular hydrogen adsorption**. These fundamentally different adsorption scenarios result in larger binding energies and are not directly comparable to molecular H_2 physisorption cases. We have clarified this distinction in the revised version of Table S1 by explicitly indicating the adsorbate type (H atom vs. H_2 molecule) and the defect structure involved.

We respectfully cite these literature values under the assumption that they were obtained using sound computational procedures unless otherwise stated by the original authors. Our intent is to illustrate the tunability of hydrogen binding strength across different BN defect configurations, not to suggest any inadequacy in prior studies.

In the manuscript "Theoretical simulations indicate that BN scaffolds with vacancies deviating from the ideal B_3N_3 structure exhibit markedly enhanced H_2 adsorption, with binding energies as low as -5.18 eV," Ignores the 3rd entry in the table - Reference S20 which presents values for H_2 adsorption on nanotubes that are much higher than previous work.

Response: Thanks for pointing out this sentence in introduction. The binding energy of -5.18 eV was extracted from Reference S21. Reference S20 (the 3rd entry in the table) indeed reported a more negative value of -5.24 eV, corresponding to the adsorption energy of **two hydrogen atoms on the BN nanotubes with defects**. The conclusion from Reference S20 is "the presence of defects on the wall of the tube enhances the adsorption of the hydrogen atoms on the tube (J. Chem. Phys. 2006,124, 054706)", which is supportive to our description.

To avoid the misunderstanding, we have revised the sentence in introduction as follows:

“For example, in perfect sp^2 -hybridized B–N hexagonal units, the binding energies of H_2 molecules were calculated to be -0.085 eV and -0.1 eV at the B and N sites, respectively. Theoretical simulations indicate that BN scaffolds with vacancies deviating from the ideal B_3N_3 structure are can activate and dissociate H_2 molecules through the formation of N–H and B–H bonds, with the adsorption energy of hydrogen atoms as low as -9.40 eV, depending on the vacancy size and atomic configuration (Supplementary Table 1).”

Shevlin et al reports Eads nanotube $V_B: -4.10$; $V_N: -0.74$

Response: Thanks for carefully reading the references. The values mentioned by the reviewer are corresponding to reaction energy (E_{reac}) instead of E_{ads} in Shevlin’s work (Reference S21: Physical Review B, 2007, 76, 024104). The correct E_{ads} values were $V_B: -4.16$ eV; $V_N: -0.80$ eV, calculated from the **atomic hydrogen adsorption on the defects of an (8,0) BN nanotube**. The adsorption energy of hydrogen molecular on defect-free BN nanotube is -0.04 eV in Reference S21, which is very close to the values reported in Reference S18 and S19. The original table in Shevlin’s work is attached as follows:

[Figure Redacted]

It is also strange that the authors choose to report H_2 Gibbs free energy of Hydrogenation from reference S23 when there are numerous studies that report the energy of adsorption.

Response: Thanks for pointing out Reference S23. We do not want to argue about the specific energy of adsorption but to compare the energy trend in different defect sites. Reference S23 (J. Phys. Chem. C 2025, 129, 14, 6657–6665) represents the most recent work that comprehensively evaluated the molecular hydrogen activation by defects in boron nitride. In addition to the energy of adsorption, Gibbs free energy of Hydrogenation also includes entropy and zero-point corrections, which is a better predictor of catalytic activity. Their conclusion “the trends for defect hydrogenation indicate that forming N–H bonds is more energetically favorable than forming B–H bonds for small size defects, while H_2 splitting by FLP would be more favorable for larger size defects.” supports our hypothesis that the hydrogen adsorption energy becomes more negative when defects were introduced.

It should also be noted that very negative binding energies will disfavor hydrogenation and favor surface hydrogenation. Binding energies should be in the range of Pt, Pd and Ni, which are around -1.6 to -0.3 eV. In this case high binding energy is not the best, one wants a moderate binding energy.

Response: We agree with the reviewer that very negative binding energies will disfavor hydrogenation and favor surface hydrogenation.

This might be the reason why the metal-free boron nitride exhibited lower activity than the noble metal catalysts. A moderate binding energy is ideal but need many efforts for the defects engineering, which is still challenging at the current stage. If the binding energy could be controlled accurately, the metal-free catalysts with a moderate binding energy hold a great potential to replace the current noble metal catalysts in industrial.

4)

Page 3 - reference S24 states that single atom defects do not activate by a FLP mechanism.

Response: Thanks for reading the reference. But there is a misunderstanding here. The focus of Reference S24 is the physical adsorption of H₂ molecules on BN. They did not discuss any catalytic systems or hydrogenation reactions. Thus, we did not claim the single substitution as the FLP mechanism. The FLP systems were summarized in Table S2 from Reference S28 to S40.

Again, the authors do not present the reference to previous work in a clear manner.
references 42-46

42- is not BN but the precursor with excess B or N

43- reports larger than single atom defects

44 - carbon scaffold with B and N functionality

45 - BCN not BN

46 - hydrogenation of olefins over defects - directly related to the presented work

Response: We appreciate the reviewer's time on carefully reading every references. Actually, our description corresponding to references 42-46 is "h-BN, as well as B- and N-doped carbon materials". The reviewer's comments validate the correct citation of these references, as they are all about h-BN, and B-/N-doped carbon materials.

Regarding the reviewer's comments about the reference 46, we have discussed this reference separately. The revised description is as follows:

“Similar mechanisms have been demonstrated in previous studies involving h-BN and B- or N-doped carbon materials.⁴²⁻⁴⁵ In addition to FLP-catalyzed hydrogenation, Nash et al. proposed that hydrogenation on defect-rich BN more closely follows the Horiuti–Polanyi mechanism.⁴⁶ This work pioneered the heterogeneous metal-free hydrogenation over defect-laden h-BN.”

Ref 46 specifically states that the mechanism is not an FLP mechanism with DFT models supporting a Horiuti–Polanyi like mechanism. This 2016 paper is corroborated by ref S24.

This work needs to be discussed in a separate sentence. Something like:

Although complete hydrogenation of olefins over defects generated in situ was reported by Nash et al[46], we achieved semi-hydrogenation through implementation of an flux reconstruction process to ...[insert insight]

Response: We respect the reviewer's comments. But we would like to express a different understanding on References 46 and S24. In reference 46 (Nash et al. ACS Omega 2016, 1, 1343–1354), Nash et al. reported the hydrogenation of olefins over defective BN in a ball mill reactor. Regarding the mechanism, Nash et al. stated “the data suggests that the hydrogenation mechanism is closer to the Horiuti–Polanyi mechanism for olefin hydrogenation over metals than that proposed for FLP catalyzed hydrogenation or acid-zeolite-catalyzed hydrogenation”. They also admitted **“It is difficult to unambiguously identify the active site and hydrogenation mechanism.”** In addition, the defects used in Nash's DFT calculations are single atom defects or substitution which were similar to the defects reported in Reference S24. However, their calculations are distinct. The hydrogen binding energy on V_B in Nash's work is -4.95 eV, likely the atomic adsorption energy. While the Reference S24, reported hydrogen molecule adsorption energy of -0.08 eV on V_B. Moreover, **Reference S24 did not discuss any catalytic systems or hydrogenation reactions.** Thus, we cannot tell “This 2016 paper is corroborated by ref S24.”

Regarding the relationship between Reference 46 and this work, we would like to point out the differences. The novelty of our work is the use of defect-rich BN for the semihydrogenation of acetylene. Acetylene is not an olefin and was not discussed in Reference 46. In addition, the defects proposed in our work (Figure 6) are not the single atom defects as proposed in Nash's work. Despite the different understanding, we appreciate and accept the reviewer's suggestion about discussing the Reference 46 in a separate sentence.

“Similar mechanisms have been demonstrated in previous studies involving h-BN and B- or N-doped carbon materials.^{42–45} In addition to FLP-catalyzed hydrogenation, Nash et al. proposed that hydrogenation on defect-rich BN more closely follows the Horiuti–Polanyi mechanism.⁴⁶ This work pioneered the heterogeneous metal-free hydrogenation over defect-laden h-BN.”

12) The point of this comment was that the authors' models indicate strong hydrogen binding to active sites. This is counter to the H₂/D₂ scrambling observed and more in line with previous theoretical calculations showing binding energies from -0.5 to -4.95. H₂/D₂ scrambling would be more plausible if binding energies were in the -2.3 to -0.3 eV range.

Response: We agree with the reviewer that H₂/D₂ scrambling would be more plausible if binding energies were in the -2.3 to -0.3 eV range, which were typically observed on noble metal catalysts. But the relatively stronger binding to active sites in BN-700 is not counter to the H₂/D₂ scrambling observed at high temperatures. This is evidenced by the observed H₂/D₂ exchange in cyclohexane at **100 °C** (Figure 4c) and in the gas phase at **250 °C** (Figures 4d and 4e). In contrast, the H₂/D₂ scrambling on noble metal catalyst was observed **at 0 °C** (Catalysis Today, 2016, 259, 9-18). The high H₂/D₂ scrambling temperature is in agree with the strong hydrogen binding to BN-700.

13) The explanation is counter the work presented in reference 46. In reference 46, propylene and cyclohexene were fully hydrogenated under fairly mild conditions. While compounds such as diphenethylene were only hydrogenated at the olefinic bond. The olefinic bond on ethene absolutely interacts with vacancies in the BN structure in a η^2 interaction. As per ref 46 ethene's binding energy on defects is on par with propene:

ethene VB: -3.71, VN -1.90

Propene VB: -3.69, VN -1.76

DFT models even show electron density sharing between the olefinic bond and the defect.

Response: Thank you for highlighting references 46.

The hydrogenation performance is dependent on the catalytic conditions and defects types in catalysts. For the catalytic conditions, the hydrogenation reaction in Reference 46 was through a mechanochemical process in a ball mill reactor. We have also reported several works on the mechanochemical synthesis (*ACS Materials Lett.* 2019, 1, 1, 83–88; *Angew. Chem. Int. Ed.* 2019, 58, 5018; *Nat. Commun.* 2020, 11, 1086; *Angew. Chem. Int. Ed.* 2020, 59, 21935). The feature of this mechanochemical process is the localized extreme high temperature and mild overall system temperature. According to the Bowden and Tabor Hot Spot Theory, postulates the existence of localized regions of high temperature, exceeding 1000°C, during the sliding of materials (<https://doi.org/10.1098/rspa.1936.0074>). Such a localized hot spot should provide enough energy to drive the fully hydrogenation of olefins as listed in Reference 46.

In contrary, our reaction was performed under a constant temperature on a fixed bed to mimic one process in the industrial catalysis and the conditions were optimized for the highly selective

semihydrogenation of acetylene. As we demonstrated in control experiments, the formation of ethane is indeed observed. For example, reducing the flow rate to 25 ccm led to an increased ethane selectivity from 3% to 7% (Supplementary Fig. 25A). In a separate experiment, increasing the catalyst loading from 75 mg to 300 mg further increased ethane selectivity to 17% (Supplementary Fig. 25B). Thus, despite the different defect structures and experimental conditions, our work does not exclude the hydrogenation of ethene on BN-700. The explanation is not counter the work presented in reference 46. It is worth to note that an optimized catalytic condition is essential to ensure the highly selective semihydrogenation of acetylene.

In the DRIFTS explanation how would acetylene hydrogen bond to a surface when all the atoms in acetylene are expected to have partial charges of 0?

Response: We thank the reviewer for the insightful question regarding acetylene adsorption on BN-700. Although acetylene is a nonpolar molecule with atoms carrying near-zero net partial charges, its acidic C–H bonds exhibit localized partial positive character on the hydrogen atoms. (Hartmann, et. al., *J. Phys. Chem A*. 105.18 (2001): 4470-4479.)

In the referenced study by Polo-Garzon et al. (*ACS Catalysis* 10.9 (2020): 5278-5287), a DRIFTS peak at $\sim 3680\text{ cm}^{-1}$ was assigned to acetylene weakly adsorbed via hydrogen bonding to form surface hydroxyl (O–H) groups. Similarly, Ivanov et al. (*Physical Chemistry Chemical Physics* 5.20 (2003): 4718-4723.) observed analogous interactions on oxide surfaces. Other spectroscopy studies and ab-initio calculations also support acetylene exhibiting weak hydrogen bonding interactions (Krajewska et. al., *Physical Chemistry Chemical Physics* 4.18 (2002): 4305-4313).

In our BN system, the DRIFTS peak at $\sim 3670\text{ cm}^{-1}$ is attributed to acetylene hydrogen bonding with surface N–H groups at edges or defect sites. These N–H groups can act as hydrogen bond donors or acceptors interacting with the partially positive C–H protons of acetylene. Thus, despite the overall neutrality of acetylene, the localized partial positive charge on the hydrogen atoms facilitates hydrogen bonding to polar N–H functionalities on BN. This weak but specific interaction perturbs the C–H stretching vibrations, producing the observed DRIFTS signature.

The persistence of this band up to $150\text{ }^\circ\text{C}$ supports the transient yet stable adsorption of acetylene via hydrogen bonding on BN surface defects. Therefore, our interpretation is consistent with both the cited literature and our spectral observations.

It is unfortunate that the key vibrational energy of the C=C stretch (1623 cm^{-1}) in ethene is mostly lost in the absorption data. Although there does seem to be a small peak shifted to lower energy (as would be expected in η^2 binding).

Response: We appreciate the reviewer's observation regarding the weak C=C stretching band in our DRIFTS spectrum. It is true that the classical C=C stretch of free ethylene occurs at $\sim 1623\text{ cm}^{-1}$, but upon binding to a surface in an η^2 (π -complex) configuration, this band typically weakens and may shift due to back-donation and weakening of the double bond.

In our spectra, we observe weak but reproducible signals at 1637 cm^{-1} , 1756 cm^{-1} , and 1798 cm^{-1} , which we associate with the formation of a weakly bound ethylene intermediate. The upward

shift and broadening of these signals are consistent with an η^2 complex and a weakening of the C=C bond upon coordination. Importantly, the absence of strong absorption in the 2900–3000 cm^{-1} region, which would typically appear if hydrogenated products were strongly anchored. This suggests that the ethylene intermediate is weakly held and desorbs quickly.

Together with the observation of N–H and B–H stretching vibrations, this supports our proposal that ethylene forms briefly on defect-rich BN sites and then desorbs, thereby limiting over hydrogenation to ethane and yielding predominantly the desirable partial hydrogenation product.

A better way to assess this binding is to saturate a catalyst sample with ethene and perform transmission FTIR in a KBR pellet mixed with a little powder. This worked well for CO₂ adsorbed on a BN surface.

Chagoya, K.L., D.J. Nash, T. Jiang, D. Le, S. Alayoglu, K.B. Idrees, X. Zhang, O.K. Farha, J.K. Harper, T.S. Rahman, and R.G. Blair, Mechanically Enhanced Catalytic Reduction of Carbon Dioxide over Defect Hexagonal Boron Nitride. ACS Sustainable Chem. Eng., 2021. 9(6): p. 2447-2455.

Response: We thank the reviewer for this thoughtful suggestion. We agree that transmission FTIR using a KBr pellet is effective for studying CO₂ adsorption on BN materials, owing to the strong interaction between CO₂ and BN surface sites, as also evidenced by our CO₂-TPD results (Figure 4a). However, this approach is less suitable for ethylene, given that BN-700 exhibits strong adsorption toward acetylene but only weak interaction with ethylene.

Moreover, transmission FTIR measurements at room temperature in a static pelletized system do not accurately reflect the dynamic conditions of catalytic hydrogenation. In contrast, our in-situ DRIFTS experiments capture surface species and reaction intermediates under flowing gas and elevated temperature conditions. These data already support our mechanistic insights that acetylene is strongly adsorbed and activated on the BN surface, while ethylene, once formed, desorbs rapidly, preventing over-hydrogenation.

Additional Comments:

A) Page 3 line 87: 14 nm crystallite sizes are a quite small and not indicative of a highly crystalline material. Especially when the region where amorphous halo for BN shows up in XRD is not in the plot.

Response: We agree. We have revised “highly crystalline” to “crystalline” here.

B) Page 5 line 153 - BN is not classically aromatic as pi electron density is localized around the nitrogen atoms.

Response: We have revised “aromatic units” as “six-membered rings”.

C) Supplemental Figure 3:A Williamson-Hall plot is produced from the integral breadths of each peak not the FWHM.

Response: Thank you for pointing this out. We have recalculated the W-H analysis using the integral breadth method and replaced the original values.

“The W-H plot and correction details are shown in Supplementary Fig. 3. The crystallite size was estimated to be 5.8 nm. The negative strain value ($\epsilon = -0.0023$) indicates compressive strain, likely due to lattice defects or dislocations, as further supported by the high dislocation density ($\delta = 2.9 \times 10^{16} \text{ m}^{-2}$).”

D) The figures in the revised manuscript are out of order.

Response: All figures have been renumbered and reorganized for proper sequential referencing.

E) In figure 5 ionothermal is misspelled.

Response: Corrected. Thank you for spotting this.

Reviewer #2 (Remarks to the Author):

The authors have made major revisions to their previous submission, which well addressed the questions raised by the reviewers and made the research work more convincing. I think it can be accepted for publication now.

Response: We appreciate the reviewer’s positive comments.

Reviewer #3 (Remarks to the Author):

This manuscript presents a rigorous and well-supported study on selective semihydrogenation of acetylene on defect-rich boron nitride catalysts, offering valuable insights. The revised paper looks very nice and the conclusions are well-justified. I recommend publication without further revision.

Response: We appreciate the reviewer’s kind words and positive comments.

Response to Reviewer's Comments

Reviewer #1 (Remarks to the Author):

The revised paper is suitable for publication. Although, I am not fully convinced by the proposed mechanism the authors do present data that suggests the rate of hydrogenation of the first pi bond in acetylene is much faster than the subsequent pi bond.

Response: We sincerely appreciate the reviewer's positive assessment of our work and value the constructive skepticism regarding the proposed mechanism. While we acknowledge that certain mechanistic aspects remain open to further validation, we believe the presented data provide strong evidence that the hydrogenation of the first π bond in acetylene proceeds significantly faster than that of the second π bond. We view this as an important direction for future in-depth mechanistic investigations.

A few comments on the authors response follow:

0) In reference to comment "Ref 46 specifically states that the mechanism is not an FLP mechanism with DFT models supporting a Horiuti–Polanyi like mechanism. This 2016 paper is corroborated by ref S24."

Correction reference S21

"We find that H₂ activation by FLP is the most favorable type of bonding only if the defect size is large enough to accommodate steric effects between H atoms"

Response: We thank the reviewer for providing the correction. Although we were unable to locate the exact quoted sentence in reference S21, its content aligns with our interpretation that H₂ activation via an FLP mechanism is favorable when defect sites in hexagonal BN are sufficiently large to accommodate steric effects between hydrogen atoms.

1) The feature of this mechanochemical process is the localized extreme high temperature and mild overall system temperature.

This is a very outdated reference and view of the mechanism of mechanochemical reactions. Much work has shown that the hot spot mechanism is highly unlikely. For example:

James, S.L., C.J. Adams, C. Bolm, D. Braga, P. Collier, T. Friščić, F. Grepioni, K.D.M. Harris, G. Hyett, W. Jones, A. Krebs, J. Mack, L. Maini, A.G. Orpen, I.P. Parkin, W.C. Shearouse, J.W. Steed, and D.C. Waddell, Mechanochemistry: opportunities for new and cleaner synthesis. Chemical Society Reviews, 2012. 41(1): p. 413-447.

<https://doi.org/10.1039/C1CS15171A>

"It seems unlikely that hot spots and magma-plasma sites are the primary sites of reactivity in molecular organic and metal–organic mechanochemical reactions. If they were, extensive decomposition would be expected. That such decomposition is not seen suggests that these phenomena may be too brief and/or too localized to be the primary reactive sites for molecular organic reactions. "

This assertion is corroborated by ref 46 where high temperatures would have resulted in severe coking - which was not evident.

A more likely explanation of the difference is that the manuscript presents work performed in a plug flow reactor whereas Nash et al reported results from a batch reactor with a large amount of catalyst (5g). In such a case complete hydrogenation would be in line with the author's observation that increases catalyst loading resulted in increased ethane yield.

Response: We thank the reviewer for highlighting the advances in understanding mechanochemical reaction mechanisms and for pointing out the limitations of the traditional “localized hot spot” model. We agree that recent studies, including the comprehensive review by James et al. (Chem. Soc. Rev. 2012, 41, 413–447), suggest that hot spots and magma–plasma sites are unlikely to be the dominant reactive environments in most molecular organic and metal–organic mechanochemical processes. In this work, we have not claimed any hot spot mechanism-related discussions.

We appreciate the reviewer’s observation regarding the differences in reactor configuration and catalyst loading between our study and that of Nash et al. In our plug-flow reactor system, the reaction conditions and catalyst bed dimensions differ significantly from the batch reactor setup with large catalyst quantities used in Ref. 46. This distinction provides a more plausible explanation for the observed differences in selectivity. The higher catalyst loadings and longer residual time of the ethene intermediate on the surface of the catalyst in batch conditions can facilitate complete hydrogenation to ethane.

2) Although there is work showing hydrogen bonding between various C-H groups with oxygen-containing species. It is not necessarily established with NH species especially given the lower bond energy that would be expected.

Response: We thank the reviewer for highlighting this point regarding hydrogen bonding between C–H groups and nitrogen-containing species. Specifically, we state in the manuscript, “Strong acetylene adsorption via hydrogen bonding to exposed nitrogen sites was also evident through persistent signals at 3254 and 3318 cm^{-1} .” While the reviewer is correct that much of the literature has focused on C–H \cdots O interactions, direct evidence for C–H \cdots N hydrogen bonding has also been well established, particularly in small-molecule model systems.

Infrared spectroscopic studies of the acetylene···ammonia complex (*Hilpert G., et al., J. Chem. Phys. 1996, 105, 6183–6191*) show a pronounced red shift of the acetylenic C–H stretch upon hydrogen-bond formation with the nitrogen lone pair, with the hydrogen-bonded C–H fundamental appearing between 3214 and 3344 cm^{-1} , compared to the monomer C–H stretch at $\sim 3289 \text{ cm}^{-1}$. This large red shift provides direct spectroscopic evidence of a C–H···N interaction.

Theoretical studies (Hartmann M., et al., *J. Phys. Chem. A. 2001, 105, 18, 4470–4479*) systematically evaluated C–H···X hydrogen bonds for acetylene, ethylene, and ethane and various hydrides, including NH_3 . The strongest hydrogen bonds were calculated for the acetylene··· NH_3 complex (9.2 kJ/mol), which is slightly stronger than the corresponding acetylene··· OH_2 complex (7.7 kJ/mol). These results indicate that acetylene can form appreciable hydrogen bonding interactions with nitrogen lone pairs, comparable to or even stronger than analogous oxygen interactions.

In our system, the defect sites on BN provide exposed nitrogen atoms with lone pairs that can act as Lewis bases, which is functionally similar to the nitrogen lone pair in NH_3 . The DRIFTS signals observed at 3254 and 3318 cm^{-1} in our work are consistent with the acetylenic C–H stretch being perturbed by such interactions, closely matching the red-shifted region reported in the acetylene··· NH_3 complexes (*Hilpert G., et al., J. Chem. Phys. 1996, 105, 6183–6191*). Taken together, both spectroscopic and computational evidence from the literature support our assignment of these persistent DRIFTS bands to acetylene adsorption via C–H···N hydrogen bonding on the exposed nitrogen sites of defect BN.